# Spatial Prediction of Groundwater Withdrawal Potential Using Shallow, Hybrid, and Deep Learning Algorithms in the Toudgha Oasis, Southeast Morocco

Lamya Ouali [1,*], Lahcen Kabiri [1], Mustapha Namous [2], Mohammed Hssaisoune [3,4], Kamal Abdelrahman [5], Mohammed S. Fnais [5], Hichame Kabiri [6], Mohammed El Hafyani [7,8], Hassane Oubaassine [9], Abdelkrim Arioua [10] and Lhoussaine Bouchaou [4,11]

1   Laboratory of Engineering Sciences and Techniques, Geo-Resource Geo-Environment Geological and Oasis Heritage Research Team, Department of Geosciences, Faculty of Sciences and Techniques, Moulay Ismail University, BP 509 Boutalamine, Errachidia 52000, Morocco
2   Laboratory of Data Science for Sustainable Earth, Sultan Moulay Slimane University, Beni Mellal 23000, Morocco
3   Laboratory of Applied Geology and Geo-Environment, Faculty of Sciences, Ibn Zohr University, Agadir 80000, Morocco
4   Faculty of Applied Sciences, Ibn Zohr University, Ait Melloul 86150, Morocco
5   Department of Geology & Geophysics, College of Science, King Saud University, P.O. Box 2455, Riyadh 11451, Saudi Arabia
6   Laboratory of Artificial Intelligence, Faculty of Sciences, Moulay Ismail University, Meknes BP11201, Morocco
7   Laboratory of Geoengineering and Environment, Research Group "Water Sciences and Environment Engineering", Department of Geology, Faculty of Sciences, Moulay Ismail University, Meknes BP11201, Morocco
8   ULiège (Gembloux Agro-Bio Tech), Terra Research Center, Water-Soil-Plant Exchange, 5030 Gembloux, Belgium
9   Laboratory of the Dynamics of the Lithosphere and the Genesis of Resources, Faculty of Sciences-Semlalia, Cadi Ayyad University, Marrakech 40000, Morocco
10  Water Resources Management and Valorization and Remote Sensing Team, Faculty of Sciences and Techniques, Sultan Moulay Slimane University, Beni Mellal 23000, Morocco
11  International Water Research Institute, Mohammed VI Polytechnic University, Ben Guerir 43150, Morocco
*   Correspondence: la.ouali@edu.umi.ac.ma; Tel.: +212-660-642-070

**Abstract:** Water availability is a key factor in territorial sustainable development. Moreover, groundwater constitutes the survival element of human life and ecosystems in arid oasis areas. Therefore, groundwater potential (GWP) identification represents a crucial step for its management and sustainable development. This study aimed to map the GWP using ten algorithms, i.e., shallow models comprising: multilayer perceptron, k-nearest neighbor, decision tree, and support vector machine algorithms; hybrid models comprising: voting, random forest, adaptive boosting, gradient boosting (GraB), and extreme gradient boosting; and the deep learning neural network. The GWP inventory map was prepared using 884 binary data, with "1" indicating a high GWP and "0" indicating an extremely low GWP. Twenty-three GWP-influencing factors have been classified into numerical data using the frequency ration method. Afterwards, they were selected based on their importance and multi-collinearity tests. The predicted GWP maps show that, on average, only 11% of the total area was predicted as a very high GWP zone and 17% and 51% were estimated as low and very low GWP zones, respectively. The performance analyses demonstrate that the applied algorithms have satisfied the validation standards for both training and validation tests with an average area under curve of 0.89 for the receiver operating characteristic. Furthermore, the models' prioritization has selected the GraB model as the outperforming algorithm for GWP mapping. This study provides decision support tools for sustainable development in an oasis area.

**Keywords:** groundwater potential; spatial prediction; machine learning; performance; water supply; oasis

## 1. Introduction

Oases in arid lands are home to thousand-year-old civilizations that have survived due to the availability of water resources and their management. Nowadays, water stress is often frequent and widespread; therefore, with the quasi-absence of surface water, groundwater represents the principal water supply source in those areas [1–5]. Thus, its availability is of paramount interest to territorial sustainable development [6]. According to the United Nations Water Conference (UNW), despite the groundwater resource abundance on Earth (90% of freshwater), data availability and specific knowledge about the resource represent a primary challenge in many countries [7]. Therefore, regular quantitative and qualitative monitoring of groundwater constitutes an essential step toward its management, which passes through its identification [8].

Consequently, mapping groundwater potential (GWP) has been one of the most crucial components of groundwater research [9]. This concept was introduced by [10] as a spatial estimate of groundwater yield capacity based on a series of indicators, i.e., GWP-influencing factors. Furthermore, it can refer to the likelihood of the availability of groundwater in a given area [11]. Previous studies have shown that common indicators such as hydrological, geological, topographical, and climatic indicators were integrated according to their influence on the GWP using several mathematical and statistical approaches. Among the latter, the frequency ratio, weights-of-evidence, and logistic regression are the approaches usually used by scholars [12–14].

In this context, the combination of remote sensing (RS) data, geographic information systems (GIS), and multi-criteria decision-making (MCDM) analyses based on statistical methods represent a popular, rapid, and cost-effective assessment method of water resource management in general and groundwater identification [3,15]. In previous years, this combination has allowed many researchers worldwide to provide large-scale spatio-temporal analyses of several datasets within various geo-environmental issues in general terms and for groundwater monitoring, particularly in [8] and the references therein. This was based on the fact that the RS data provide rapid and replicated observations on environmental characterization and monitoring; the GIS offers various data spatial analysis and visualization tools, whereas the MCDM analysis combines several quantitative and qualitative factors to solve complex spatial problems in order to develop decision support tools [16–18]. Nonetheless, in oasis areas, GWP identification using the cited methods is rarely applied.

Recently, numerous novel methods and algorithms related to artificial intelligence (AI) based on machine learning (ML) and deep learning (DL) have been developed, assessed, and approved in the field of GWP mapping determination; this has been conducted with respect to inventories of water withdrawal points and geological, hydrogeological, hydrological, topographic and climatic factors [11,17,19–21]. On this matter, the following models were commonly used and applied in the sub-cited studies: random forest (RF), support vector machine (SVM), linear regression (LR), decision tree (DT), naive Bayes (NB), convolutional neural network (CNN), long short-term memory (LSTM) and artificial neural network (ANN) [6,9,14,22–25]. Furthermore, a variety of methods have also been proposed to improve the efficiency and precision of the prediction models, such as optimization algorithms and ensemble models [23,26,27].

In the present research, ten groundwater potential prediction models were applied, evaluated, and compared, including using different types of existing algorithms [28]: shallow models, i.e., traditional ML models, hybrid models (bagging, boosting, ensemble), and deep learning models. The used models were: multilayer perceptron (MLP), k-nearest neighbor (KNN), DT, SVM, voting, RF, adaptive boosting (AdB), gradient boosting (GraB), eXtreme gradient boosting (XGraB), and deep learning neural network (DLNN). These algorithms were selected due to them being the most commonly used individual models; this allowed us to compare our findings with previous results demonstrated in different environments as investigated by [13,20,23,29], etc. Furthermore, they are the most powerful boosting and deep learning models, and we used the models in order to overcome the

research gap related to the comparison and validation of different types of GWP modeling algorithms.

The Toudgha Oasis is commonly known for its environmental and socio-ecological sites' importance, and its economy is essentially based on rural and subsistence agriculture, tourism, and mining activities, which are directly related to water availability. The area belongs to the oases of southern Morocco, which have been declared by the United Nations Educational, Scientific and Cultural Organization (UNESCO) as a 'Biosphere Reserve' (www.unesco.org accessed on 20 June 2022). Furthermore, it houses several territories of life that can be defined as Indigenous peoples and local Communities Conserved Areas and territories (ICCAs) [30], where several local communities survive because of the groundwater resources, which have been exploited by ancient hydraulic systems such as khettarats (i.e., Qantas). the latter are the essential techniques that have been developed to guarantee the oasis' sustainability, especially in the face of climate change [31]. Overall, the identification of the GWP in the Toudgha Oasis is essential for the development of sustainable strategies for groundwater exploitation, protection, and management, and thus the preservation of this valuable ecosystem.

Due to the unavailability of previous research on GWP, the above-mentioned approach is unprecedented in the chosen study area and has not been assessed before in a similar oasis region. Furthermore, in this work, the GWP mapping includes, in particular, the application of ML and DL models using Google Colaboratory as a Web Integrated Development Environment (WIDE) based on a total of 884 input data and twenty-three GWP-influencing factors, which were mapped using RS data and the GIS environment. The main purposes of this study were first to select the key influencing factors related to the GWP, produce predicted GWP maps, and evaluate the prediction models' performance and prioritization. Overall, this research deals with the elaboration of advancing and cost-effective groundwater resource planning and management tools in the Toudgha Oasis.

## 2. Materials and Methods

### 2.1. The Study Area

The study was conducted within the Toudgha watershed, which is in the southeast of the Moroccan kingdom and covers an area of 2296 km$^2$, ranging between 31°51′35″ N–31°9′14″ N and 5°10′45″ W–5°57′01″ W (Figure 1). The area is characterized by an arid climate [3],which is strongly influenced by the continental and desert context. The interannual average rainfall is 154 mm, which shows high spatial and temporal variability marked by a spatial difference in the average precipitation of 110 mm between the northwest (i.e., the mountainous region) and the southeast (i.e., the lowland region) and a temporal difference in the average precipitation of 295 mm between the wettest and driest years. The daily thermal difference may reach 22 °C, with a monthly average temperature ranging between 8 °C and 27.8 °C.

Geologically, the area is localized at a threefold junction between the Central High Atlas Mountain (Jurassic) in the North, the Eastern Anti-Atlas Mountain (Paleozoic and Precambrian) in the South, and the pre-African furrow (Cretaceous) in the middle, shaped as a dissymmetric synclinorium where most of the Paleogene–Pliocene and Quaternary sediments have been deposited (Figure 2a). The Precambrian presents the outcrops of intrusive and metasediment formation; the Paleozoic is materialized by sandstone and shale bundled in a sedimentary sequence dating from the lower Cambrian to the Carboniferous [32]. The Jurassic rocks are composed of limestone and dolostone (Liassic–Dogger) [33], whereas the Cretaceous is mostly represented by sandstone, conglomerates (Infra-Cenomanian), and limestone (Cenomanian–Turonian and Senonian) [34,35]. Finally, the Paleogene–Pliocene succession is constituted by polygenic conglomerates. The quaternary is formed by alluvial, silt, sand, and conglomerates. The geological structure of the area (Figure 2b,c) is dominantly affected by a fractured system belonging to the South Atlasic Fault showing an overall NE direction between the High Atlas Mountain and the pre-African furrow [36,37]. The cross-section (Figure 2c) shows that the deformation in the study area is characterized

by the presence of folds and thrusts, which are more developed close to the South Atlas Fault Zone; it decreases in the southeast part, resulting in more spaced open syncline and anticline folds [36].

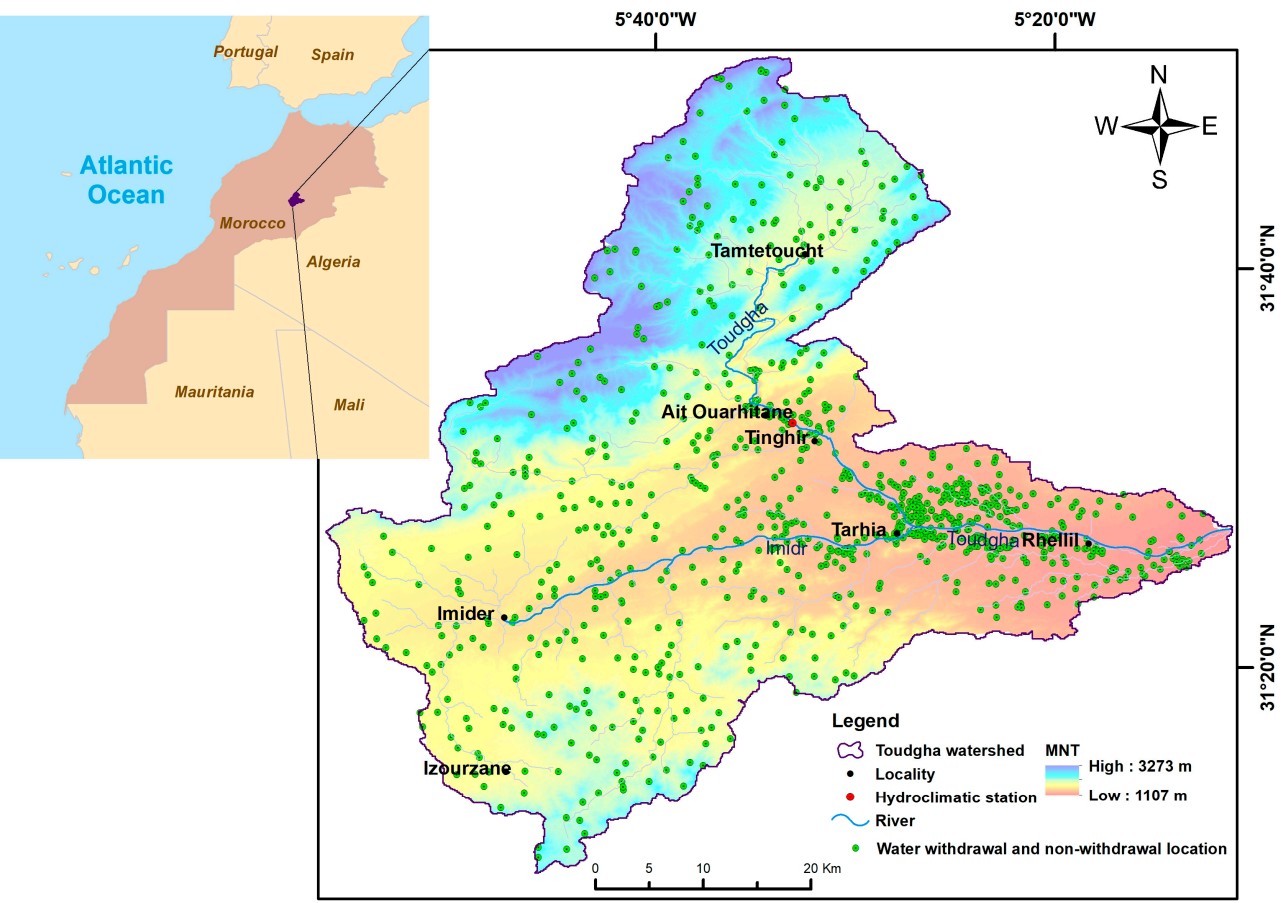

**Figure 1.** Geographic placement of the study area.

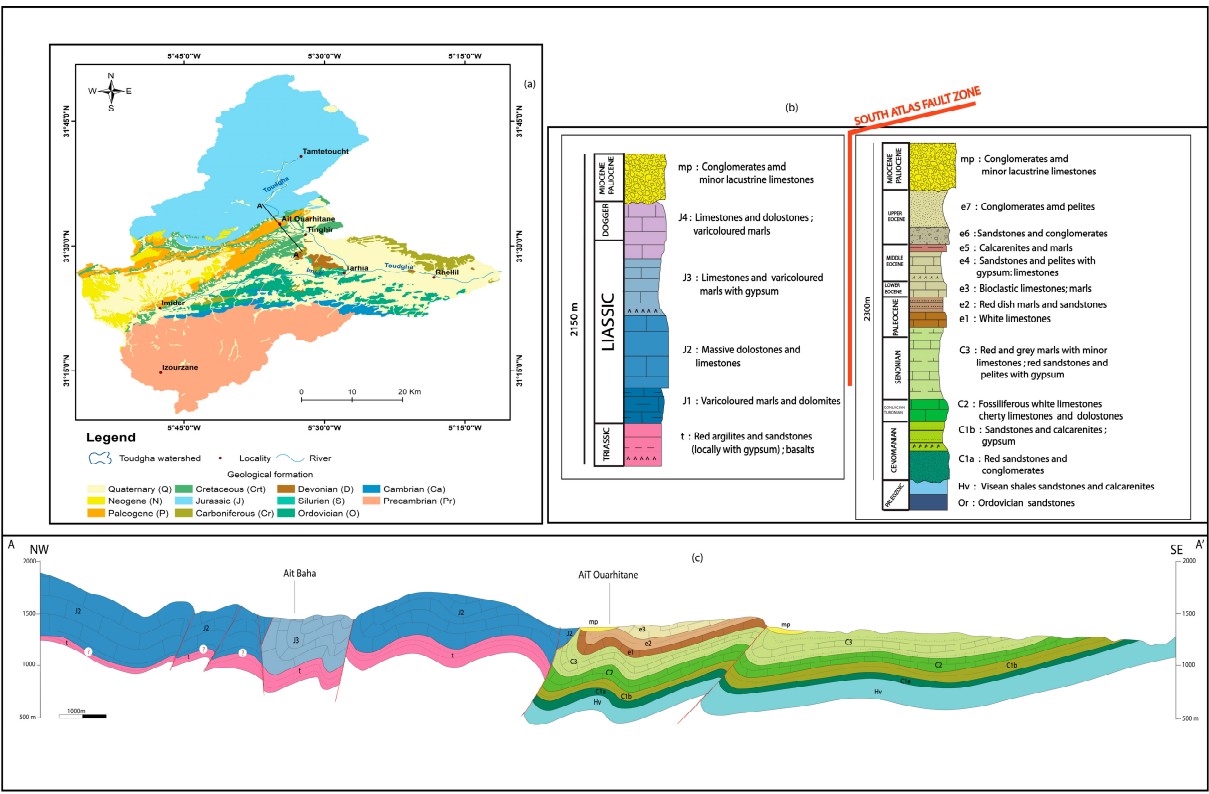

**Figure 2.** Geological setting of the study area. (**a**) Simplified geological map; (**b**) high and Anti-Atlas stratigraphic logs (in [36]); (**c**) geological cross-section (in [36]).

The groundwater resources are distributed in four different hydrogeological systems of variable importance [38–42], which are arranged in juxtaposition from north to south. According to the technical data sheet analyses of the inventoried water withdrawal points, we distinguish: (1) The High Atlas deep aquifer, which is fractured and often karstic, contains both a Liassic level and Dogger level, corresponds to dolomitic limestone and marly limestone, respectively, and characterized by higher productivity (exceed 100 L/s). These aquifers provide several springs, among which the most famous in the study area are the Toudgha gorges springs and the Sacred Fishes spring in the downstream part of the Toudgha gorges; (2) the Cretaceous deep aquifer, which includes two main levels: the Infra-Cenomanian sandstones, which is the most productive level at the area (about 14 L/s), and the Senonian sands and sandstones; (3) the Anti-Atlas deep aquifer, which contains limited resources (productivity average less than 1 L/s) circulating in Ordovician sandstone and Devonian limestone; (4) the Toudgha alluvium aquifer, generally oriented along the Toudgha river in which the water flows in the quaternary alluvial. Fluvial-lacustrine formations are characterized by an extreme permeability. Groundwater flow is very discontinuous in the northwest, where it is strongly controlled by the faults. However, it is homogeneous in the southeastern region, where it is controlled by the characteristics of the different formations.

### 2.2. Methodology and Data Sources

The adopted methodology of this study is summarized in Figure 3, which describes the three major steps. Step (I) is devoted to the input database generation, which is composed of the GWP inventory map and the GWP-influencing factors maps, namely, geological factors (lithology, fracture density (FD), closeness to fracture (CF)), climatic factor (Rainfall), land surface factors (Normalize Difference Vegetation Index (NDVI), LULC), hydrological factors (drainage density (DD), runoff depth (Q), closeness to stream (CS) (i.e., distance to stream), Stream Power Index (SPI), Topographic Wetness Index (TWI)), and

topographic factors (aspect, convergence, curvature, elevation, slope length (LS), Melton ruggedness number (MRN), multi-resolution ridge top flatness (MRRTF), multi-resolution valley bottom flatness (MRVBF), plan curvature, profile curvature, slope, and Terrain Ruggedness Index (TRI)). The input database was transferred into numerical data using the frequency ration (FR) method in order to determine the relationships between the GWP and the influencing factors. Then, the GWP-influencing factors were selected according to their importance and multi-collinearity analyses such as correlation matrix (CM) results, variance inflation factors (*VIF*), tolerances (Tol), and mutual information (*MI*) test. Step (II) was focused on the models' application and produced the predicted GWP maps. Step (III) was dedicated to assessing the efficiency and the ranking of the prediction models according to several validation criteria related to performance metrics such as sensitivity (Se), specificity (Sp), precision (Pr), false positive rate (FPR), accuracy (Ac), F1-score, mean absolute error (MAE), root-mean-square error (RMSE), and the area under curve of the receiver operating characteristic (AUC-ROC). In the second and third steps, the database was randomly divided into training data and testing data used for the generation of the GWP maps and for evaluating the medals' performances, respectively. A preliminary stability test of the prediction and success rates regarding optimal data partitioning was performed.

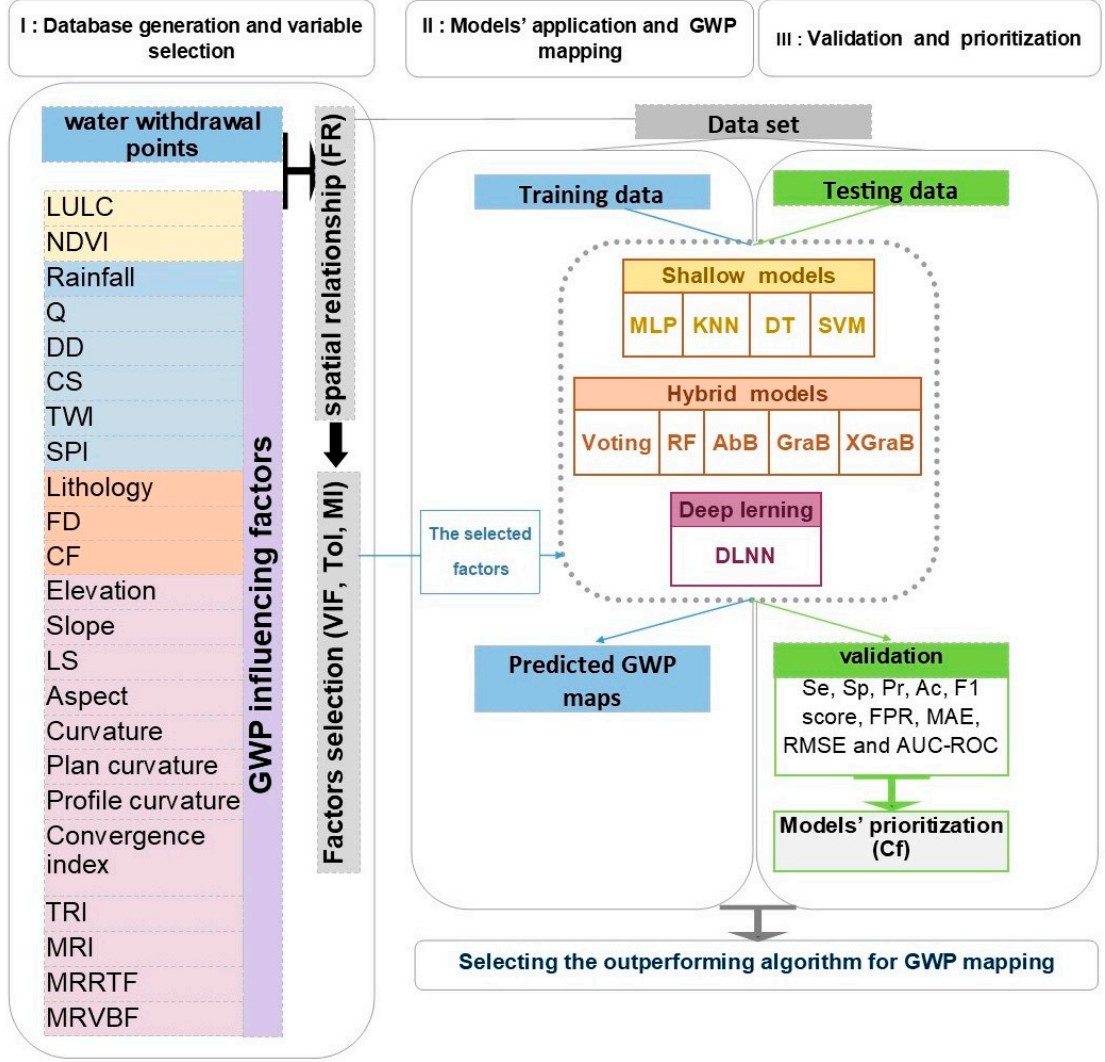

**Figure 3.** Flowchart of the adopted methodology.

　　Table 1 shows the various sources of the acquired data, while the details of the method used are outlined in the next sections.

**Table 1.** Database and sources used.

| Data | | Source |
|---|---|---|
| Water withdrawal points inventory | | Field investigation<br>Data collection from:<br>the Hydraulic Agency of Guir, Ziz, Rheris Basin (ABHGZR)<br>the Regional Office of Agricultural Development (ORMVAO)<br>the National Office of Electricity and Drinking Water (ONEE) |
| GPM_3IMERGM v06 produced by the Global Precipitation Measurement (GPM) mission [43]. Pixel size: 10 km × 10 km | | https://giovanni.gsfc.nasa.gov/giovanni/<br>(Accessed on 1 July 2022) |
| Digital Elevation Model (DEM) Produced by ALOS PALSAR.<br>Pixel size: 12.5 m × 12.5 m | | https://vertex.daac.asf.alaska.edu/<br>(Accessed on 1 July 2022) |
| Landsat Oli-8 images.<br>Pixel size: 30 m × 30 m | | https://earthexplorer.usgs.gov/<br>(Accessed on 1 July 2022) |
| Geological Maps.<br>Scale: 1/200,000 | Jbel Saghro-Dades | Geological service of Morocco |
| | Toudgha Maider | |
| | Ouaouizaght Dades | |
| | Haut Atlas of Midelt | |

*2.3. Database Generation*

2.3.1. Water Withdrawal Points Inventorying

The water withdrawal inventory is an indispensable component of water resources management. Therefore, it presents a baseline step in the GWP mapping [14]. The study area is known for several types of groundwater withdrawal systems, and its inventorying was prepared after extensive field investigation and data collection.

The data related to the exploratory and monitoring drillings and wells were collected from the ABHGZR. The irrigation water exploitation data for the irrigation was gathered from the ORMVAO. The data related to the drinking water exploitation within the urban area were collected from the ONEE, whereas the drinking water exploitation data within the rural area, the inventorying of springs, and the Khettarats (i.e., Qantas) were collected through field visits. The GWP of the water withdrawal systems was determined from available pumping tests, organizations' reports (ABHGZR, ORMVAO, and ONEE), and field surveys with the local community. Accordingly, a high GWP was assigned to high productive withdrawal at a yield value $\geq 86.4$ m$^3$/d. Therefore, 442 water withdrawal points have been identified.

Several scholars recommend integrating sampling data from areas without groundwater withdrawal in order to balance the input dataset ([8,13,20] and references therein; therefore, the list of the water withdrawal systems (set to 1, i.e., a high GWP) inventoried was completed using the same number of locations with no water withdrawal systems (set to 0, i.e., a very low GWP), and it was randomly mapped using the function create random points in a GIS environment, resulting in a total of 884 points (Figure 1). At this step, a variation in the percentage of the data split (training/testing) has been noted among the most recent studies, in which the division of 70/30% was the most used partition [14,23,25] in addition to the partition 75/25% and 80/20%, which have been used by Namous et al. [20] and Talukdar et al. [9], respectively. Accordingly, the 884 points were randomly divided into training (70%) and validation (30%) data, which were used to develop and validate the groundwater potential prediction models. Furthermore, to evaluate the stability of the models related to the partitioning data, the prelaminar test values of the prediction and success rates using the three cited partitions (70/30%; 75/25%, and 80/20%) were examined.

2.3.2. Mapping the GWP-Influencing Factor

The GWP is dependent upon several varied factors. In this regard, Díaz-Alcaide and Martínez-Santos [10] have confirmed through a systematic review that twenty factors were

found to be commonly integrated, of which geology, lineaments, landforms, land use/land cover (LULC), rainfall distribution, drainage density, and slope were mostly always present. However, the determination of the variables influencing the GWP is still strongly controlled by data availability. Accordingly, and based on the previous studies [15,23] twenty-three factors were considered in the present work, including geological factors (lithology, FD, CF), climatic factor (rainfall), land surface factors (NDVI, LULC), hydrological factors (DD, Q, CS, SPI, TWI), and topographic factors (aspect, convergence, curvature, elevation, LS, MRN, MRRTF, MRVBF, plan curvature, profile curvature, slope, and TRI). In this regard, the above-mentioned factors have been identified considering that: first, the geological factors presents a direct influence on groundwater storage conditions and water quality [15]; second, the climatic factor is a crucial component in the groundwater recharge process as it controls the water resource availability [44]; third, the land surface factors that influence water infiltration and surface runoff have a substantial impact on groundwater potential and the recharge process [45]; finally, the hydrologic and topographic factors are essential for determining hydrological conditions such as groundwater flow and soil moisture [46].

The GWP-influencing factors maps preparation was achieved through distinctive primary (RS data) and secondary (published data) data sources, which have been processed using numerous spatial analysis initiatives provided by the GIS environment.

The land surface factors maps and fracture maps (includes lineament, i.e., faults and fractures) have been generated from Landsat OLI-8 images with supervised classifications to map the LULC: image analysis to map the NDVI; principal component analysis; and lineament automatic extraction to produce the base layer to map the FD and CF. Extensive interpretation using a geologic map, topographic map, high-resolution images from Google Earth Pro, and the colored composition of the Landsat OLI-8 image bands was carried out in order to sort and eliminate lineaments other than fractures (roads, rivers, etc.). The climatic factor map was obtained from the Global Precipitation Measurement (GPM) data validated by Ouali et al. [3] within the studied area. The lithology map was digitized from the 1:200,000 geological map of Morocco. From the DEM data, topographic and hydrologic factors were derived using the terrain analysis tools except for the Q map, which has been elaborated by integrating the LULC map, hydrologic soil group map, and rainfall map on the basis of the CN-SCS method [47], adapted by the Food and Agriculture Organization (FAO) for application in North Africa [48]. Considering that the majority (fourteen factors) of the integrated factors present a spatial resolution of 12.5 m × 12.5 m, all other thematic maps have been resized to the same level using a resampling method. Therefore, a total of twenty-three maps were prepared (Figure 4).

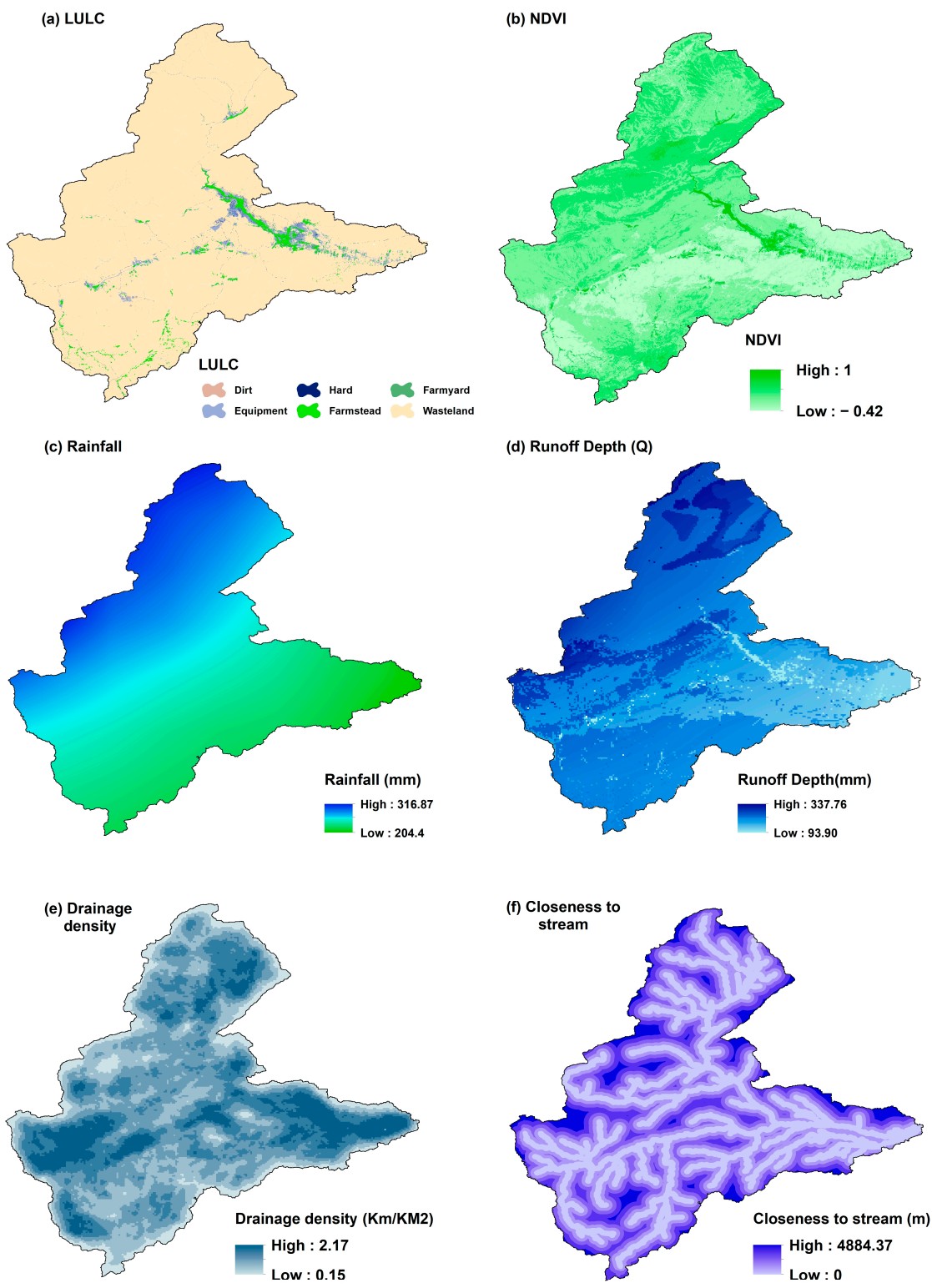

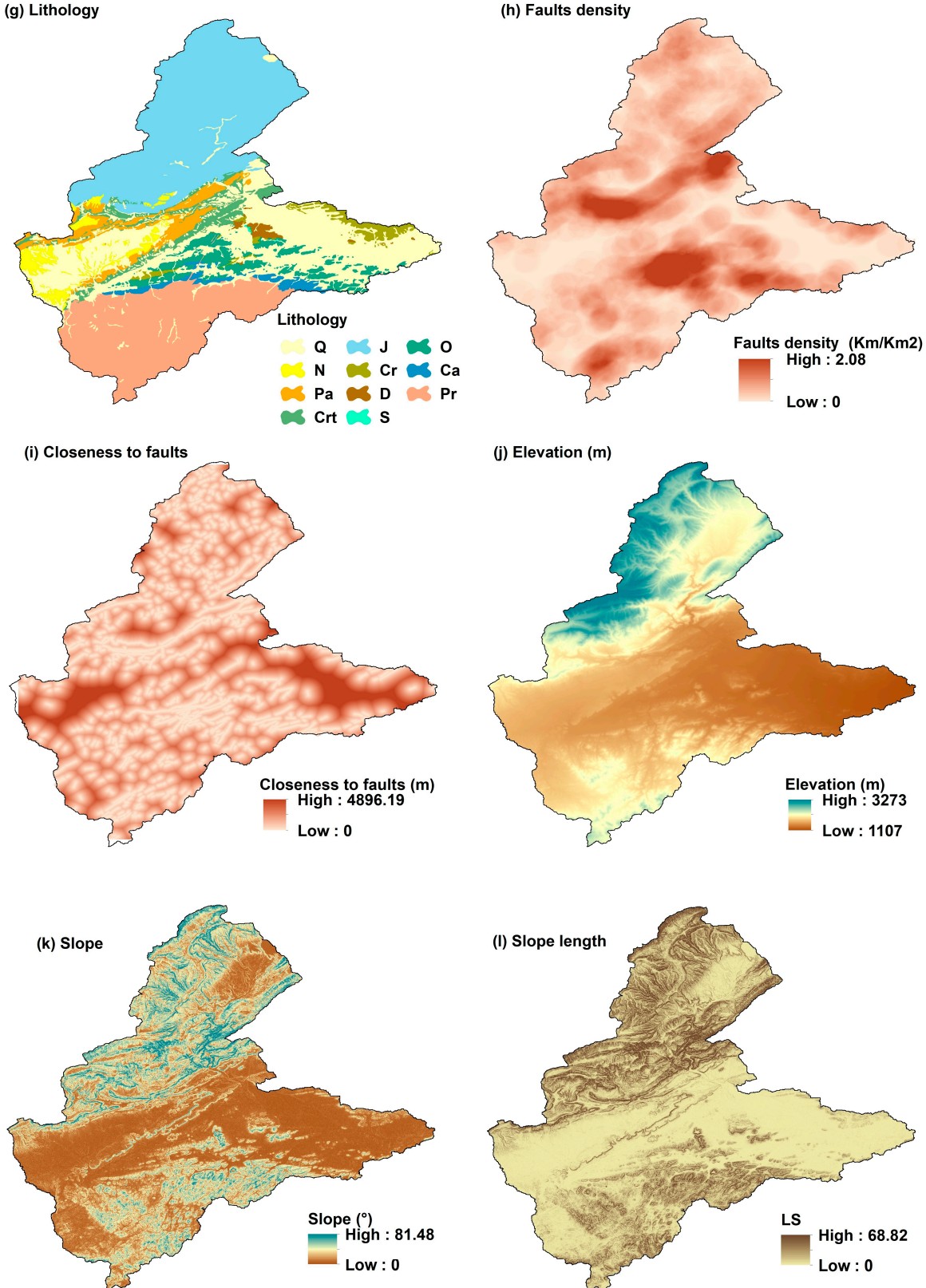

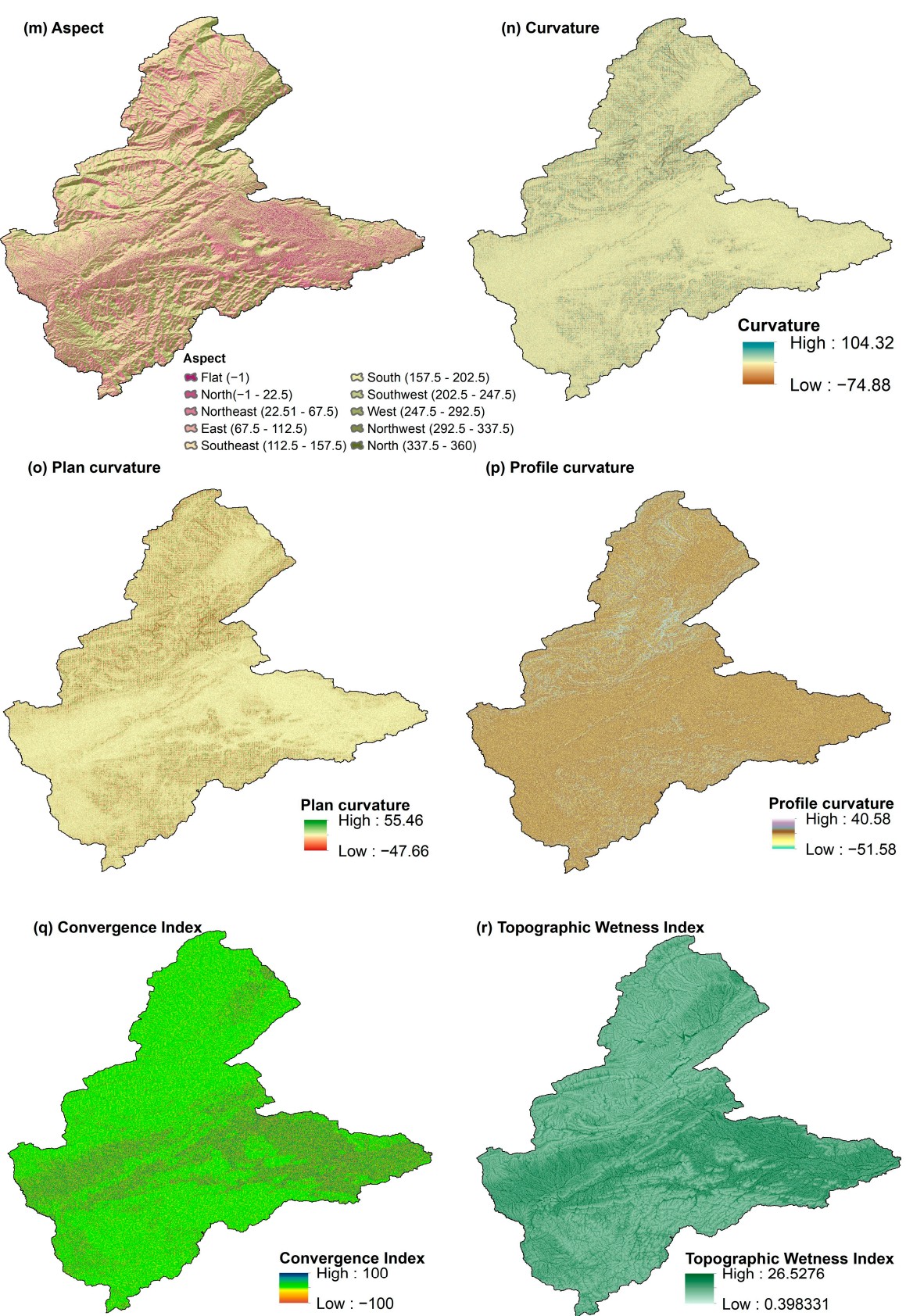

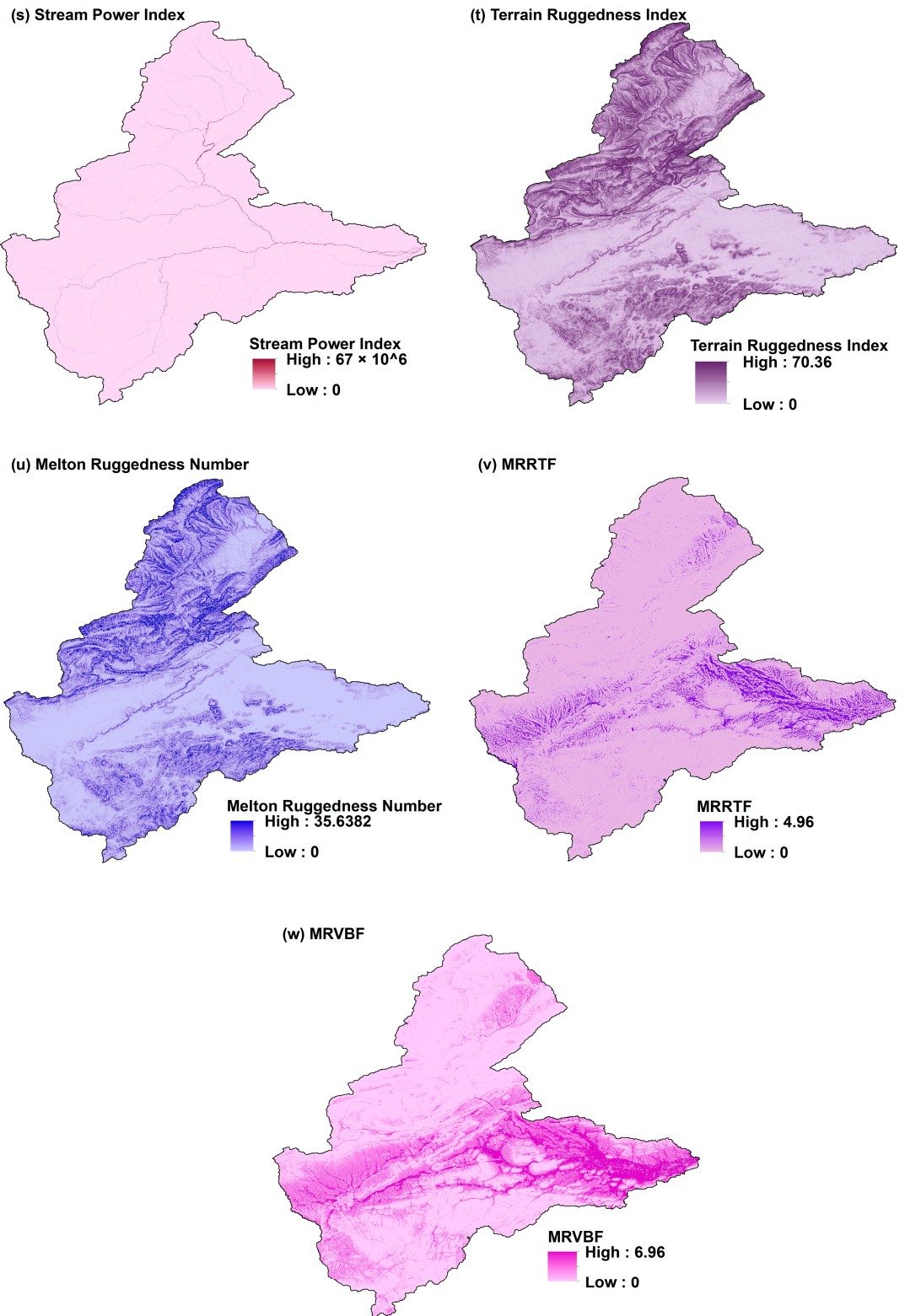

**Figure 4.** Maps of the groundwater potential-influencing factors. (**a**) LULC; (**b**) NDVI; (**c**) rainfall; (**d**) runoff depth; (**e**) drainage density; (**f**) closeness to stream; (**g**) lithology; (**h**) faults density; (**i**) closeness to faults; (**j**) elevation; (**k**) slope; (**l**) slope length; (**m**) aspect; (**n**) curvature; (**o**) plan curvature; (**p**) profile curvature; (**q**) Convergence Index; (**r**) Topographic Wetness Index; (**s**) Stream Power Index; (**t**) Terrain Ruggedness Index; (**u**) Melton ruggedness number; (**v**) multi-resolution ridge top flatness; (**w**) multi-resolution valley bottom flatness.

### 2.3.3. GWP-Influencing Factors Optimized Selection Analysis

The GWP-influencing factors selection optimization analysis reduces the prediction models' complexity, and it aims to select the most appropriate factors [9,23]. In this study, to advance the ML GWP prediction, the integrated GWP-influencing factors have undergone several primary tests, including the multicollinearity analyses of CM, *VIF* (Equation (1)), *Tol* (Equation (2)), and *MI* (Equation (3)). It allows for the elimination of redundant (e.g., collinear factors) and insignificant factors. According to O'Brien [49] and Zhou et al. [50], the high multicollinearity is indicated by a *Tol* score of 0.1 and a *VIF* value of >10. Furthermore, according to the CM for each highly correlated pair of factors that respond to these requirements, whichever one has the highest *VIF* is to be eliminated. *MI* provides information about the GWP-influencing factors' importance. A negative value of *MI* indicates that the factor has no effect and will be eliminated.

$$Tol\ j = 1 - R_j^2 \tag{1}$$

$$VIF\ j = \left\lceil \frac{1}{Tol j} \right\rceil \tag{2}$$

$$MI(n, j) = H(n) - H(n/j) \tag{3}$$

where *j* is the GWP-influencing factor, *n* is the subclass of the GWP-influencing factors, *Tol i* is the tolerance of *j*, *VIF j* is the variance inflation factors of *j*, *MI* (*n*; *j*) is the mutual information for *n* and *j*, *R* is the determinant coefficient of the regression for predisposing of j, on all the other predisposing factors, *H*(*n*) is the entropy of *n*, and *H* (*n* | *j*) is the conditional entropy for *n* given the groundwater condition factor *j*.

The GWP-influencing factors optimized selection analysis and models application were founded on the normalized frequency ratio (*NFR*) determination, which has recently been a recommended step for unifying the input data type importance to the varied factors [20,51]. Therefore, the frequency ratio (*FR*) (Equation (4)) was assigned for the GWP-influencing factors' subclass in the sense of defining the relationship between the water withdrawal locations (i.e., GWP) and the feature (i.e., GWP-influencing factors) [6]. Then, the results were normalized using Equation (5). Accordingly, all of the used maps have been converted to an *NFR* ranging between 0 (low GWP) and 1 (high GWP).

$$FRn = \frac{\frac{Wn}{Wt}}{\frac{Pn}{Pt}} \tag{4}$$

$$NFRn = \frac{FRn - Max(FRn)}{Max(FRn) - Min(FRn)} * (0.99 - 0.01) + 0.01 \tag{5}$$

where *n* is the subclass of the GWP-influencing factors, *FRn* is frequency ratio of *n*, *NFRn* is the normalized frequency ratio of *n*, *Wn* is the number of water withdrawal points localized in the *n*, *Wt* is the total number of water withdrawal points, *Pn* is the number of pixels of *n*, and *Pt* is the total number of all pixels.

The GWP-influencing factors' subclass have been determined through the classification of the produced maps using the Jenks natural break technique [52]; the exceptions are the aspect, LULC, and lithology, which have been accordingly classed depending upon the directional units, supervised classification, and lithological units, respectively.

### 2.4. GWP Prediction Algorithms

In this research, ten algorithms were employed (MLP, KNN, DT, SVM, RF, AdB, GraB, XGraB, DLNN, and voting) to predict the GWP. For the description, Table 2 illustrates the selected algorithms. Furthermore, details about their classification, functionality, and parameters are available in Liu and Lang [28] and Sarker [53].

**Table 2.** Description of the applied algorithms.

| Model | Developed by | Description | Applied by |
|---|---|---|---|
| MLP | Rosenblatt [54] and Rumelhart et al. [55] | Multilayer perceptron is a fully connected, feedforward artificial neural network (ANN) comprising several nodes, including three layers, as the input layer (i.e., GWP-influencing factor), a hidden layer (i.e., weights application), and an output layer (i.e., GWP). | Farooq et al. [56] and Kanj et al. [57] |
| KNN | Fix and Hodges [58] and Cover and Hart [59] | K-nearest neighbor is founded on the collector theory. The prediction of a new data point (i.e., GWP) is based on the simple majority vote of its nearest real data points (i.e., water withdrawal points and non- water withdrawal points). | Kombo et al. [60] and Aburub et al. [61] |
| DT (CART) | Breiman et al. [62] | Decision tree is structured like a tree, where the algorithm selects the most suitable features (i.e., GWP-influencing factor) as a root and generates the child nodes. The prediction is based on top-down observations and processing results at each level, from the rote to the child nodes corresponding to the new data (i.e., GWP). | Zhao et al. [63] and Choubin et al. [64] |
| SVM | Vapnik [65,66] | Support vector machine is a non-parametric kernel-based model aimed at locating a maximum margin separation hyperplane in the n-dimensional feature space. The radial basis function was used to predict the GWP. | Liu et al. [67] and Ijlil et al. [68] |
| Voting | Littlestone and Warmuth [69] | Defined as a hybrid model, it is based on an aggregation approach that combines the results of several algorithms. In this study, it was used for the shallow (i.e., DT, SVM, MLP, and KNN) models to balance out their individual weaknesses. | Saqlain et al. [70] and Gandhi and Pandey [16] |
| RF | Breiman et al. [71] | Random forest is a meta-estimator that adjusts a given number of DT classifiers on various subsets of the GWP-influencing factors obtained by the bootstrap aggregation (i.e., bagging) and random feature selection methods. It serves to optimize the predictive efficiency of the model. | Das and Saha [23], Liu et al. [14] |
| AdB | Freund and Schapire [72] | Adaptive boosting is an ensemble learning algorithm; it is founded on combining several basic and weak predictors to produce more effective trees. It tweaks the instance weights at every interaction. | Mosavi et al. [26] |
| GraB | Ridgeway [73] | Gradient Boosting is also known as gradient tree boosting, and its approach is like the AdB algorithm. It serves to minimizes the overall prediction error (i.e., the loss function). | Park and Kim [13] |
| XGraB | Chen and Guestrin [74] | eXtreme Gradient Boosting is an improved GraB algorithm with a structure offering parallel tree boosting. It employs second-order derivatives that reduce the loss function and provide more accurate trees. | Naghibi et al. [21], Park and Kim [13] |
| DLNN | Kingma and Ba [75] | Deep learning neural network is an advanced ANN that consists of several layers: the input layer, several hidden layers, and the output layer. | Pradhan et al. [76] |

*2.5. Performance Analysis and Model's Reinking*

In the present study, the prediction models' performance was evaluated based on several indicators, such as Se, Sp, Pr, FPR, Ac, F1-score, MAE, RMSE, and AUC-ROC. These indicators are determined using Equation (6)–(14). Higher values of *Se*, *Sp*, *Pr*, *Ac*, F1 *score*, and *AUC-ROC* are proportional to good performance. In contrast, higher values of FPR, MAE, and RMSE indicate lower performance [23].

$$Se = \frac{TP}{TP + FN} \tag{6}$$

$$Sp = \frac{TN}{FN + TN} \tag{7}$$

$$Pr = \frac{TP}{TP + FP} \tag{8}$$

$$FPR = \frac{FP}{FP + TN} \tag{9}$$

$$Ac = \frac{TN + TP}{TP + FP + TN + TP} \tag{10}$$

$$F1\ score = \frac{2}{\frac{1}{Pr} + \frac{1}{Recal}}\ Recall = \frac{TP}{TP + FN} \tag{11}$$

$$MAE = \frac{1}{m}\sum_{i=1}^{m}\left|\left(Vi_{\ predicted} - Vi_{\ real}\right)\right| \tag{12}$$

$$RMSE = \sqrt{\frac{1}{m}\sum_{i=1}^{m}\left(Vi_{\ predicted} - Vi_{\ real}\right)^2} \tag{13}$$

$$AUC = \frac{\sum TP + \sum TN}{P + N} \tag{14}$$

where *TP* is the true positive, *TN* is the true negative, *FP* is the false positive, *FN* is the false negative, *P* is positive, and *N* is negative. $Vi_{\ predicted}$ is the predicted classes of GWP, $Vi_{\ real}$ is the real class of the GWP of the tested model, and *m* is the total number of predicted and real values.

The prediction models' prioritization was applied to select the most efficient model using the compound factor (*Cf*). The method is based on the ranking of the performance indicators calculated for each model in the previous section according to Equation (15).

$$Cf = \frac{1}{n}\sum_{i=1}^{n} R \tag{15}$$

where *R* is the rank of the model and *n* is the indicator.

## 3. Results

### 3.1. GWP-Influencing Factors Selection

The results of the multicollinearity analysis show a Tol value that ranges between 0.95 and 0.095 for the SPI and the TRI, respectively, as well as a *VIF* maximum value of 10.5 for the TRI and a minimum value of 1.06 for the SPI (Figure 5a). The CM depicted a strong linear correlation between the following factors: elevation with the runoff, the LS with the slope, and the TRI with the LS, where the highest *VIFs* among each pair are presented for the TRI, the Ls, and the elevation (Figure 5b). Afterwards, and according to the Tol and *VIF* requirements cited above, among the twenty-three factors, the TRI, the LS, and the slope were removed in the following analysis. After that, the *MI* of the other twenty factors (Figure 6) presents positive values that range from 0.207 (lithology) to 0.0005 (FD). Therefore, lithology is classified as the most important, followed by the Q (*MI* = 0.201) and the rainfall (*MI* = 0.106).

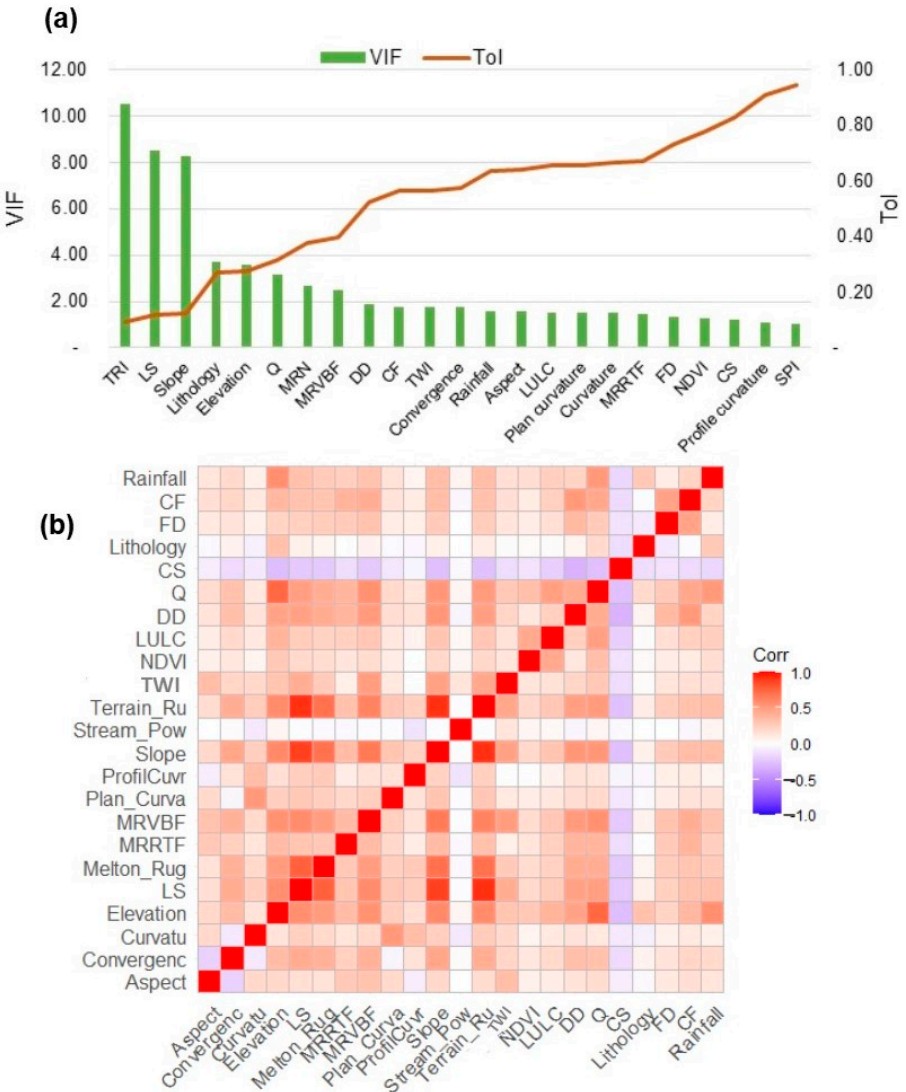

**Figure 5.** Multicollinearity analysis of the GWP-influencing factors. (**a**) Tol and *VIF*; (**b**) the correlation matrix.

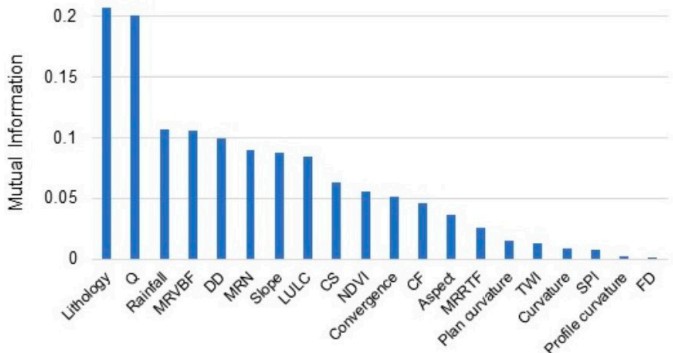

**Figure 6.** Mutual information of the GWP-influencing factors.

### 3.2. Groundwater Potentiality Maps

The GWP was mapped based on the application of the ten algorithms. The results were presented as a predicted probability ranging between 0 and 1, corresponding to the lower and higher GWP, respectively. The produced maps were then classified into the

following five different zones using the Jenks natural break classification: very low, low, moderate, high, and very high.

The first visual analysis of the ten resulting maps produced by the MLP (Figure 7a), KNN (Figure 7b), DT (Figure 7c), SVM (Figure 7d), voting (Figure 7e), RF (Figure 7f), AdB (Figure 7g), GraB (Figure 7h), XGraB (Figure 7i), and DLNN (Figure 7j) models shows that the very high GWP values are concentrated at the eastern part, particularly in the plain zone, and they are slightly represented in the western part. Meanwhile, the very low GWP values are localized in the northern and the southern part.

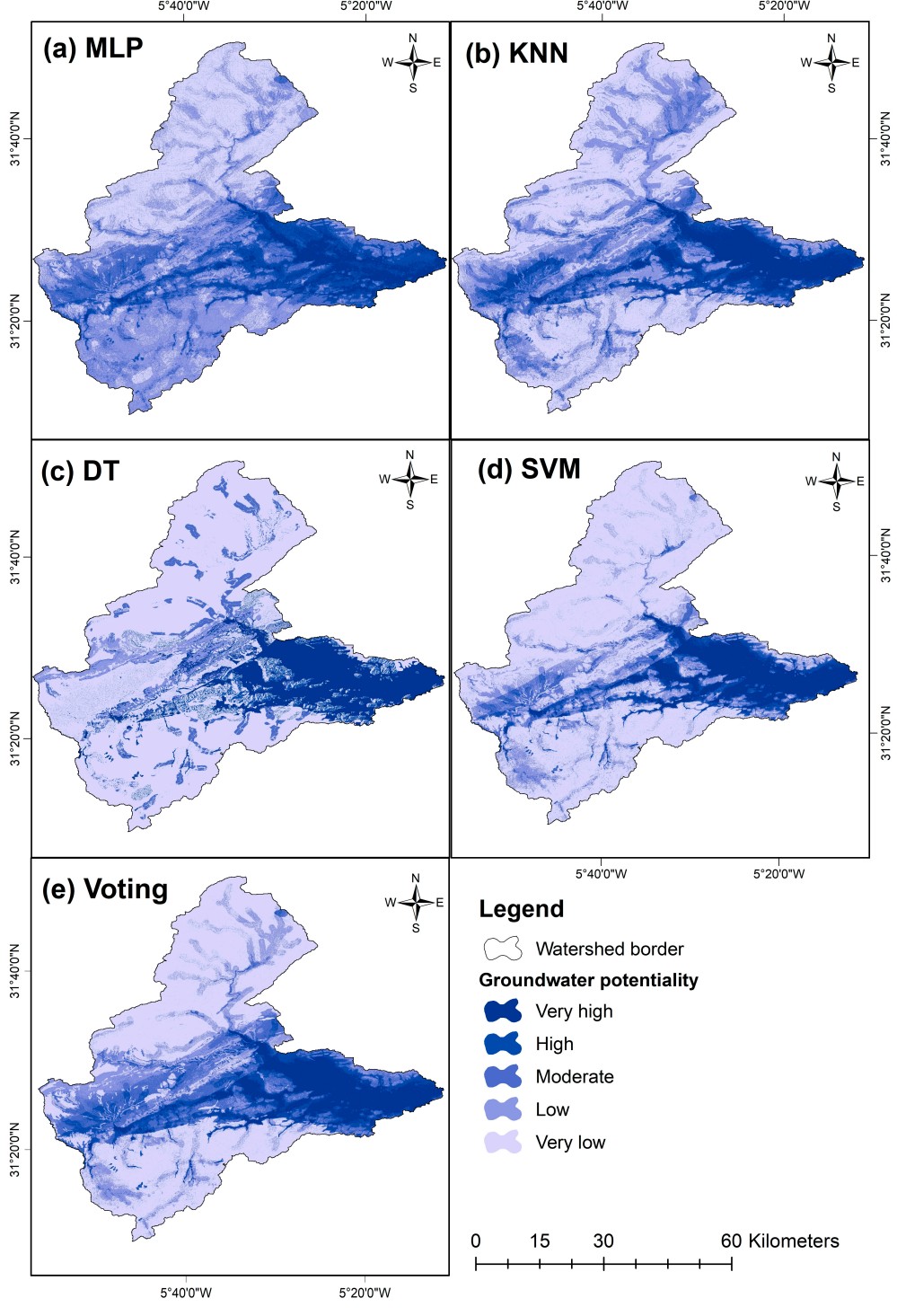

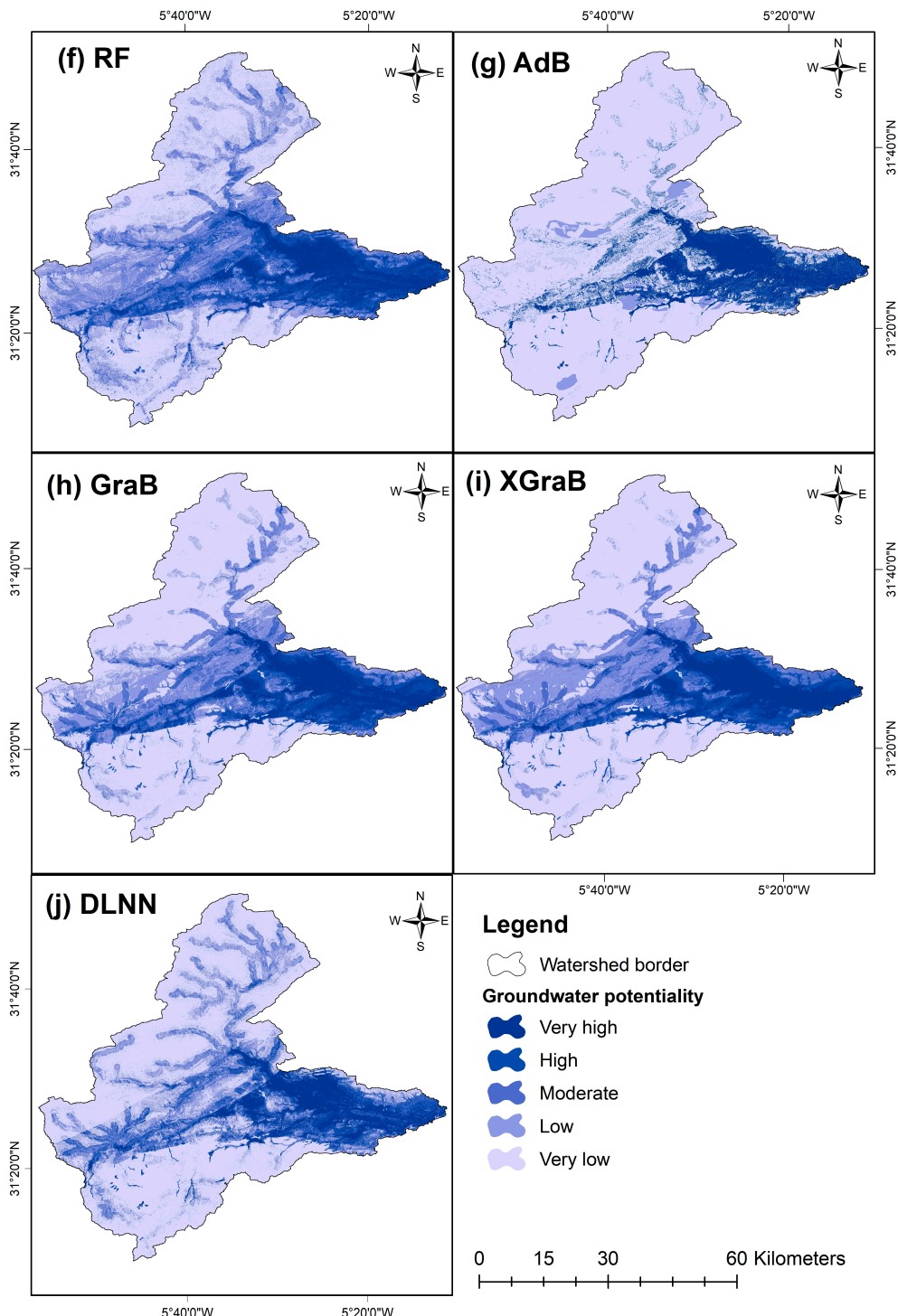

**Figure 7.** GWP maps predicted by (**a**) MLP; (**b**) KNN; (**c**) DT; (**d**) SVM; (**e**) voting; (**f**) RF; (**g**) AdB; (**h**) GraB; (**i**) XGraB; and (**j**) DLNN models.

For the shallow models, the results show that the very low, low, moderate, high, and very high GWP covers 28%, 34%, 19%, 13%, and 7% of the study area, respectively, for the MLP model (Figures 7a and 8)), 38%, 28%, 15%, 9%, and 10% of the study area, respectively, for the KNN model (Figures 7b and 8), 66%, 7%, 6%, 3%, and 18% of the study area, respectively, for the DT model (Figures 7c and 8), and 60%, 15%, 8%, 6%, and 12% of the study area, respectively, for the SVM model (Figures 7d and 8).

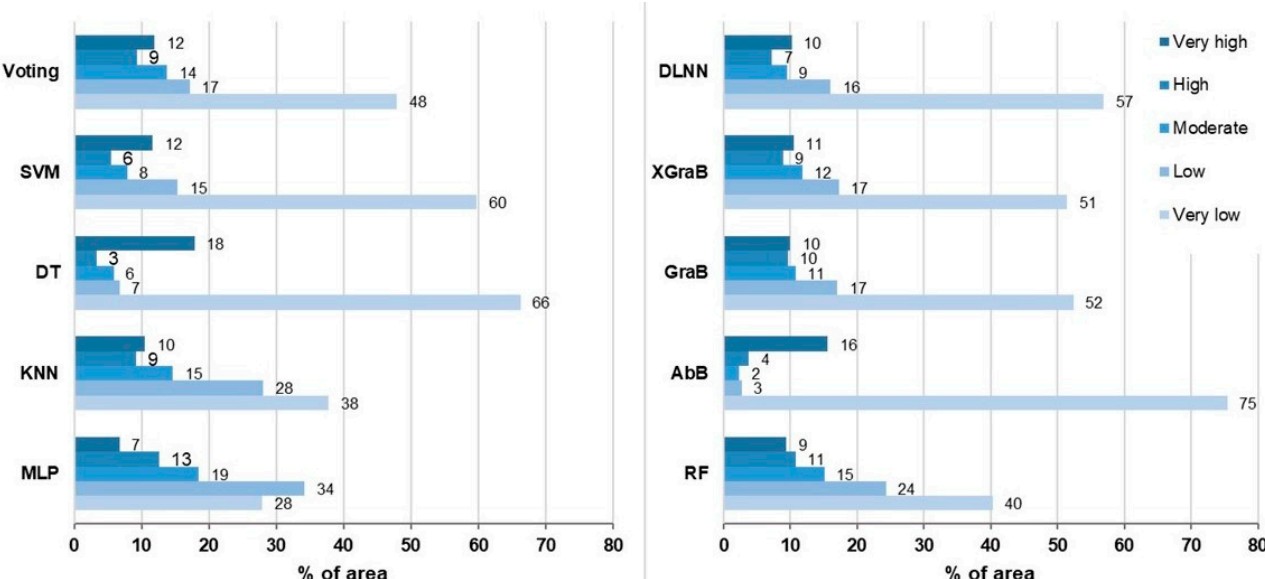

**Figure 8.** Percentage of the area occupied by the different GWP classes of the used models.

In the case of the hybrid models, the five GWP classes (very low, low, moderate, high, and very high) cover 48%, 17%, 14%, 9%, and 12% of the areas, respectively, for the voting model (Figures 7e and 8), 40%, 24%, 15%, 11%, and 9% of the areas, respectively, for the RF (Figures 7f and 8) model, 75%, 3%, 2%, 4%, and 26% of the areas, respectively, for the AdB model (Figures 7g and 8), 52%, 17%, 11%, 10%, and 10% of the areas, respectively, for the GraB model (Figures 7h and 8), and 51%, 17%, 12%, 9%, and 11% of the areas, respectively for the XGraB model (Figures 7i and 8) respectively. For the deep learning model (i.e., DLNN), most areas are predicted as having very low (57%) and low (16%) GWP, and the remaining area is associated with moderate (9%), high (7%), and very high (10%) GWP values (Figures 7j and 8).

In general, the proportion of the GWP classes (Figure 8) shows that the low and very low GWP are the dominating classes; they cover more than 60% of the total area, except for the GWP map predicted by the AdB and DT algorithm, in which only the very low GWP class dominates. However, the last presented category is the high GWP class with an average percentage of 8% of the total area. The moderate and the very high GWP classes simultaneously present 11% on average for the entire area.

*3.3. Models' Performance*

The success rate was determined using the training dataset, which reflects the accuracy of the model fit to the observed GWP. The prediction rate was determined using testing data, which shows how well the model predicts the GWP.

The success and prediction and rates of the applied models were examined for the training/testing partition at ratios of 70/30%, 75/25%, and 80/20% of the dataset; the results are shown in Figure 9. Considering the success rate for the three used partitions, the results show equal values for the RF (success rate = 1) and equal values (variation average of 0.02) over other models except for the MLP, which presents a slight variability. On the other hand, the prediction rate presents a slight variation between the results of the three partitions for each model. Nevertheless, a high prediction rate is noted for the GraB (0.83) models, followed by the AdB (0.82) and XGraB (0.82) models for the 70/30% partition.

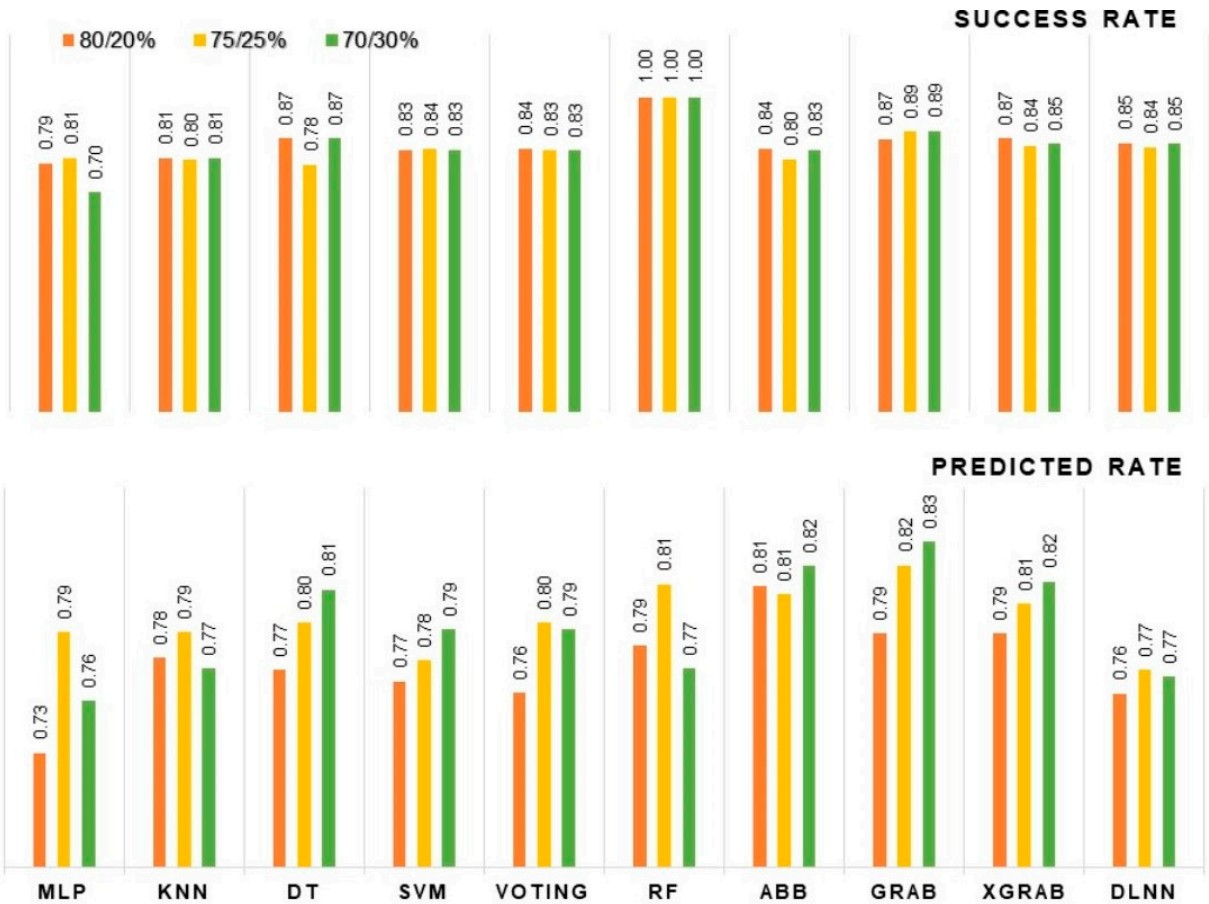

**Figure 9.** The success and prediction rates of the used models using 70/30%, 75/25%, and 80/20% partitions.

To further assess the ten models' efficiencies at predicting the GWP, the Se, Sp, Pr, Ac, F1-score, FPR, MAE, RMSE, and AUC-ROC were evaluated using the training (70%) and testing dataset (30%). The results of this part are presented in Table 3 and Figure 10.

**Table 3.** Validation indicators of the used models using the training data and validation data.

| Model | Performance Indicators | | | | | | | | |
|---|---|---|---|---|---|---|---|---|---|
| | Se | Sp | Pr | FPR | Ac | F1-Score | MAE | RMSE | AUC |
| Training Data | | | | | | | | | |
| RF | 1.00 | 1.00 | 1.00 | 0.00 | 1.00 | 1.00 | 0.00 | 0.00 | 1.00 |
| MLP | 0.83 | 0.70 | 0.70 | 0.30 | 0.76 | 0.76 | 0.24 | 0.24 | 0.85 |
| GraB | 0.91 | 0.87 | 0.86 | 0.13 | 0.89 | 0.88 | 0.11 | 0.33 | 0.96 |
| AdB | 0.84 | 0.83 | 0.80 | 0.17 | 0.83 | 0.82 | 0.17 | 0.41 | 0.91 |
| DT | 0.89 | 0.85 | 0.83 | 0.15 | 0.87 | 0.86 | 0.13 | 0.36 | 0.92 |
| SVM | 0.81 | 0.82 | 0.79 | 0.18 | 0.82 | 0.80 | 0.18 | 0.43 | 0.88 |
| KNN | 0.76 | 0.84 | 0.80 | 0.16 | 0.81 | 0.78 | 0.19 | 0.44 | 0.88 |
| XGraB | 0.86 | 0.84 | 0.82 | 0.16 | 0.85 | 0.84 | 0.15 | 0.38 | 0.91 |
| Voting | 0.84 | 0.82 | 0.79 | 0.18 | 0.83 | 0.81 | 0.17 | 0.42 | 0.95 |
| DLNN | 0.88 | 0.88 | 0.86 | 0.12 | 0.88 | 0.87 | 0.12 | 0.35 | 0.89 |

**Table 3.** *Cont*.

| Model | Performance Indicators | | | | | | | | |
|---|---|---|---|---|---|---|---|---|---|
| | **Se** | **Sp** | **Pr** | **FPR** | **Ac** | **F1-Score** | **MAE** | **RMSE** | **AUC** |
| | **Testing Data** | | | | | | | | |
| RF | 0.83 | 0.79 | 0.78 | 0.21 | 0.80 | 0.80 | 0.20 | 0.44 | 0.89 |
| MLP | 0.81 | 0.64 | 0.67 | 0.36 | 0.72 | 0.73 | 0.28 | 0.53 | 0.85 |
| GraB | 0.87 | 0.80 | 0.80 | 0.20 | 0.83 | 0.83 | 0.17 | 0.41 | 0.91 |
| AdB | 0.87 | 0.78 | 0.78 | 0.22 | 0.82 | 0.82 | 0.18 | 0.42 | 0.89 |
| DT | 0.83 | 0.78 | 0.77 | 0.22 | 0.80 | 0.80 | 0.20 | 0.44 | 0.81 |
| SVM | 0.80 | 0.77 | 0.76 | 0.23 | 0.79 | 0.78 | 0.21 | 0.46 | 0.88 |
| KNN | 0.77 | 0.78 | 0.76 | 0.22 | 0.77 | 0.76 | 0.23 | 0.47 | 0.87 |
| XGraB | 0.86 | 0.78 | 0.78 | 0.22 | 0.82 | 0.82 | 0.18 | 0.43 | 0.94 |
| Voting | 0.80 | 0.76 | 0.75 | 0.24 | 0.78 | 0.77 | 0.22 | 0.47 | 0.88 |
| DLNN | 0.81 | 0.76 | 0.76 | 0.24 | 0.79 | 0.78 | 0.21 | 0.46 | 0.88 |

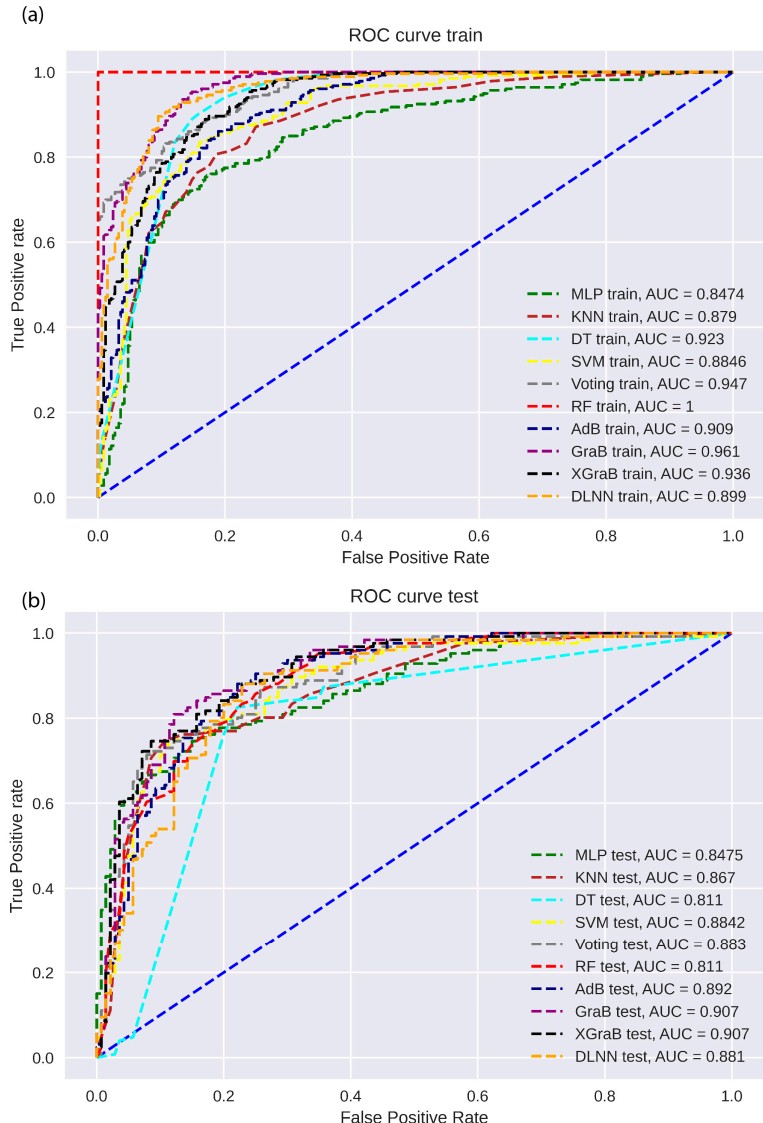

**Figure 10.** The receiver operating characteristic (ROC) curves. (**a**) The success rate using the training data. (**b**) The prediction rate using the testing data.

For the training dataset, the RF model shows an excellent and flawless prediction, presented by maximum Se, Sp, Pr, Ac, AUC, and F1-scores (values = 1) and null FPR, MAE, and RMSE scores (values = 0). The GraB, AdB, DT, XGraB, and DLNN models show very high performance, indicated by Se, Sp, Pr, Ac, AUC, and F1-score values being above 0.80 and negligible FPR, MAE, and RMSE scores. For the MLP, SVM, KNN, and voting models, the validation indicators show a high performance as presented by a minimum match value of 0.7 (i.e., for values of Se, Sp, Pr, Ac, AUC, and F1-score) and a maximum gaps value of 0.44 (i.e., for values of FPR, MAE, and RMSE).

For the testing dataset, all of the applied models present a high to very high performance. The Se values range from 0.87 for the AdB and 0.77 for the KNN, the Sp, Pr, Ac, F1-score, FPR, MAE, and RMSE values are between 0.8 and 0.64, 0.8 and 0.67, 0.83 and 0.73, 0.83 and 0.73, 0.20 and 0.36, 0.17 and 0.28 and, 0.41 and 0.53, respectively, the maximum values are all marked for the GraB model, and the minimal values are all marked for the MLP model. The AUC-ROC ranges from 0.81 for the DT to 0.91 for the GraB.

### 3.4. Models' Prioritization

Considering the variation in the models' rankings relative to each performance metric, the highest-performing model was selected using the Cf (Figure 11). The model prioritization results indicate that, for the training dataset, the RF model is the best GWP predictive model followed by the GraB model. In the case of the testing dataset, GraB and AdB are the highest-performing models for the GWP prediction. In contrast, the MLP model is classified as the lowest performing model for both cases.

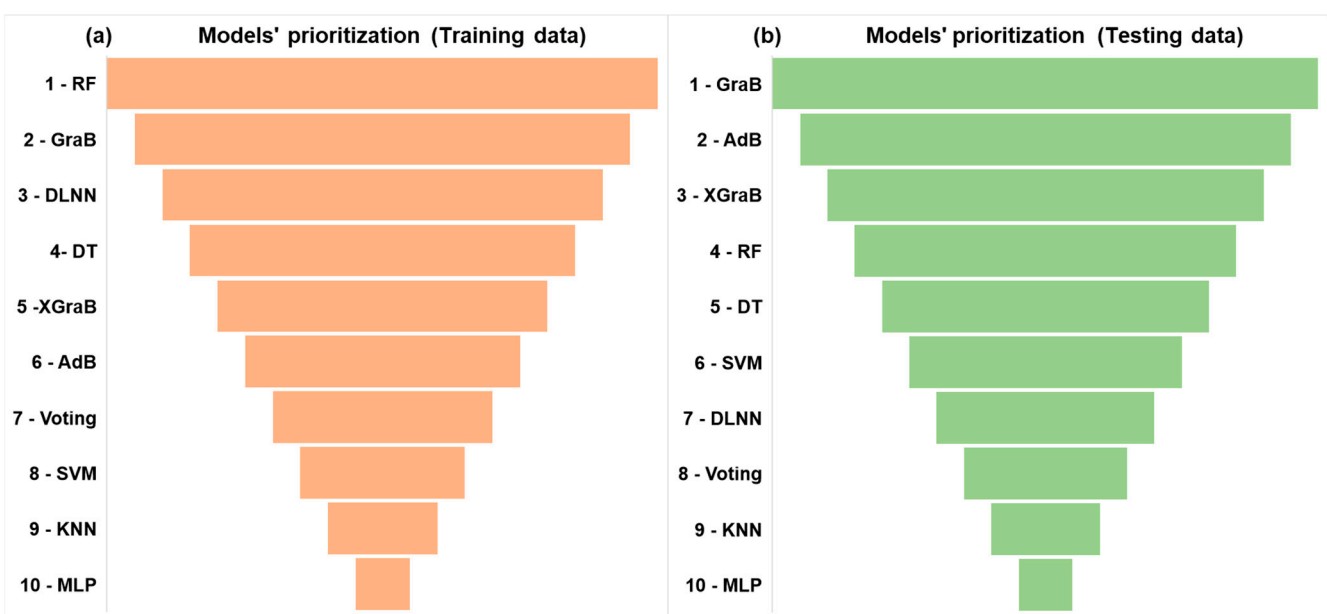

**Figure 11.** Models' prioritization based on the compound factor. (**a**) In the training step; (**b**) in the validation step.

## 4. Discussion

In recent years, the combined effect of the rapidly increasing water needs and the prolongation of drought periods requires effective and timely interventions that ensure the sustainable planning and management of water resources. Nowadays, researchers have experienced the benefits of AI technology, particularly in GWP mapping in different areas over the world using several ML and DL algorithms and numerous influencing geo-environmental factors. In this regard, due to the models' variety, shallow models, a bagging model, boosting models, a deep learning model, and a hybrid model were applied, evaluated, and compared to provide the most efficient GWP map for its management.

This section discusses in detail (i) the GWP-influencing factors and their importance, (ii) the models' performance and periodization, and (iii) the achievement of the used methodology.

### 4.1. Importance of GWP-Influencing Factors

After the water withdrawal points' field investigation and data collection, the first step for GWP spatial prediction was the influencing factors dataset preparation. Amongst the twenty-three prepared factors, the TRI, LS, and elevation have been eliminated according to their collinearity with other factors that limits the prediction performance. On the other hand, according to the *MI*, the most crucial factor is lithology, which is in line with the results of Namous et al. [20], followed by the Q. Meanwhile the less key factor is the FD. Additionally, rainfall, MRVBR, DD, slope, LULC, CS, NDVI, convergence, CF, aspect, MRRTF, plan curvature, TWI, curvature, SPI, and profile curvature have presented a considerable impact on the GWP, respectively. The importance accorded to lithology was confirmed by several researchers given that geology has a major influence on the water mobilization process; it directly controls groundwater recharge and storage of an area depending on several criteria related to lithology, faulting, and karstification. In fact, areas with higher porosity (rocks permeability and/or fractures) permit water infiltration and provide pathways for water to flow into the subsurface [15]. Furthermore, it was described by Díaz-Alcaide and Martínez-Santos [10] as the single most crucial factor for determining groundwater occurrence. The Q was integrated for the first time on the GWP prediction analysis. Indeed, it presents a high importance that confirms the interdependence of groundwater and surface water [3]. In contrast to previous research, the FD was classified as the less important factor. Considering the geomorphology of the studied area, the findings are also in accordance with the predicted GWP maps (Figure 7). In fact, the high and very high GWP classes are almost localized everywhere that the formations are permeable, especially the alluvium, sand, and sandstone formation, which are characterized by a thickness that can reach 30 m. Meanwhile, karstic limestone and the fissured rocks are characterized by a high DF and therefore low GWP; this can be justified by the fact that the fractured formation, especially in high reliefs, can be considered as groundwater recharge areas. Accordingly, we fervently request further studies concerning groundwater flow origin of recharge in the Toudgha Oasis.

### 4.2. Models' Performance and Periodization

The examination of the training and validation datasets splitting indicates considerable stability for the 70/30% division. In fact, it represents an essential part of selecting the most adequate partition to achieve high performance. The models' evaluation indicates that all of the applied algorithms have a high to very high efficiency for GWP prediction, despite a slight weakness being noted for the MLP. MLP's weak performance is also reported by previous studies in different fields, such as in the spatial prediction of a landslide hazard by Hong et al. [44], in lithofacies prediction by Nwaila et al. [77], in groundwater level prediction by Kombo et al. [60], in groundwater pollution risk mapping by Ijlil et al. [68], etc.

According to the success rate, the RF model performed excellently, as is the usual in other studies on city and urban scales [13] and at a large mountainous scale [20], followed by the GraB model, which has outperformed the XGraB, unlike in the result of Park and Kim (2021). Besides these points, the GraB and AdB show the most accurate prediction rates. The low performance of other models can be directly related to the high value of the tested prediction error indicators, such as the FPR, MAE, and RMSE. However, the spatial variability of the erroneous predictions shows a clustered pattern and positive spatial correlation in the prediction errors for all of the used models. Furthermore, according to the Cf, the RF and GraB models are ranked as the most efficient models for the testing and the training datasets, respectively. However, The RF rank has been lowered for the testing dataset because of a slight decrease in the performance parameter values and an

increase in the error indicators; this instability of the RF model has also been confirmed by previous studies [23,78]. the DLNN is performs slightly poorer, and this result can be related to the fact that deep learning algorithms require more training data than other machine learning models [79]. According to the result, the GraB model is the most efficient and stable model for the GWP spatial prediction in the present study. The performance of the GraB model was also confirmed by Sahour et al. [29] for predicting the groundwater salinity in a coastal aquifer. Overall, the boosted models (GraB, AdB, and XGraB) and bagged model (RF) have presented more efficiency and stability than the deep learning models, shallow models (i.e., MLP, DT, KNN, SVM), and their ensemble (i.e., voting). This requires us to disregard shallow models and their ensembles and to use more boosted models. Therefore, we strongly recommend future research to include ensemble-boosted models in order to improve their performance.

*4.3. Effectiveness of the Used Methodology*

Regarding the used methodology, this work is in line with recent research [22,80,81] that has highlighted the importance of ML-DL, RS, and GIS in spatial decision-making, especially in the field of groundwater management, on account of their high and rapid efficiency. It responds to the major importance of water resources management in arid oasis areas with financial limitations and those undergoing socioeconomic growth. Furthermore, the resulting tools conform to the application of the sixth and the thirteenth sustainable development goals, which aim to guarantee sustainable water resources management and climate change effect mitigation.

The GWP map predicted by the GraB algorithm has excellent performance and may be used for groundwater management in the Toudgha Oasis; essentially, it can be used for locating water monitoring points, i.e., piezometers, and can map the pumping prohibition zones upstream of the Khettarats and springs. In addition, this map can be included with the MCDM analysis in order to identify the potential area for artificial groundwater recharge and/or the development of underground dams.

The achievement of this method depends on several parameters related to the input dataset, as well as on the algorithm's application and validation. Regarding the input dataset, this work has integrated the maximum of the available spatial dataset of the study area in addition to an extensive field investigation of the groundwater mobilization systems inventory. Otherwise, the application and comparison of ten diverse models based on the same dataset have provided a more reasonable model prioritization.

Therefore, as a limitation of this method, the resulting GWP maps may be changed for every dataset integration and/or the application of novel models. Otherwise, the method has focused more on the quantitative assessment of GWP; to this end, we recommend that future research conducts assessments of this methodology in other areas with different geo-environmental characteristics, with the integration of qualitative indicators, in order to guarantee both water availability and usefulness.

## 5. Conclusions

This research adopts an alternative and efficient approach to select the outperforming spatial GWP prediction model for an arid oasis area. In fact, distinct types of algorithms (MLP, KNN, DT, SVM, voting, RF, AdB, GraB, XGraB, and DLNN) have been applied, evaluated, and compared based on 442 inventoried water withdrawal points and twenty-three geo-environmental factors. Among the latter, the lithology and Q factors have been designated as the key influencing factors for the GWP, while the TRI, LS, and elevation were eliminated from the analysis according to the multicollinearity test and *MI* importance. The GWP spatial distribution demonstrates an approximately similar variation for the ten produced maps, which have been confirmed by the geomorphological particularity of the study area. Furthermore, the validation of the GWP maps was simultaneously tested with the model's evaluation using nine performance metrics (Se, Sp, Pr, Ac, AUC-ROC, F1-score, FPR, MAE, and RMSE), where the findings depicted that all of the applied models have

satisfied the validation standards for both the training and validation tests of the GWP prediction. Nevertheless, water management needs to be based on the most advanced and powerful techniques. Therefore, the models' prioritization analysis has been applied. The results demonstrate the outperforming of the GraB model (AUC training = 0.90, AUC testing = 0.96) and accordingly the efficiency of the GWP map as predicted by this model. The methodology developed in this study has generated a necessary ML-RS-GIS-based tool for GWP mapping devoted to implementing suitable water resources protection and management plans in arid oasis areas, and it could be adapted and applied in other regions around the world.

**Author Contributions:** Conceptualization, L.O., L.K., M.N. and M.H.; methodology, L.O. and M.N.; software, L.O., H.K. and H.O.; validation, L.O. and M.N.; formal analysis, L.O., H.K. and H.O.; investigation, L.O.; resources, L.K.; data curation, L.O.; writing—original draft preparation, L.O.; writing—review and editing, L.K., M.N., M.H., M.E.H., A.A., K.A., M.S.F. and L.B.; visualization, L.O.; supervision, L.K., M.N. and M.H.; project administration, L.O. All authors have read and agreed to the published version of the manuscript.

**Funding:** This research received no external funding.

**Institutional Review Board Statement:** Not applicable.

**Informed Consent Statement:** Not applicable.

**Data Availability Statement:** The data supporting the reported results of this study are available on request from the corresponding author.

**Acknowledgments:** Deep thanks and gratitude to the Researchers Supporting Project number (RSP2023R351), King Saud University, Riyadh, Saudi Arabia for supporting this research article. The authors are grateful to the ABHGZR, ORMVAO, and ONEE for the supply of the necessary data, and they also wish to thank the three anonymous reviewers for their constructive comments.

**Conflicts of Interest:** The authors declare no conflict of interest.

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
