# Peer review of "Spatial Prediction of Groundwater Withdrawal Potential Using Shallow, Hybrid, and Deep Learning Algorithms in the Toudgha Oasis, Southeast Morocco"

_sustainability, doi:10.3390/su15053874_

Round 1
Reviewer 1 Report
Dear authors,
In this manuscript, you compare 10 methods to estimate groundwater potential maps of the Toudgha Oasis. Globally, the manuscript is rather well presented and pleasant to read, except for the cross-referencing or hyperlink errors. However, some important aspects are missing: notably a good review of previous hydrogeological studies at this site (and how do your work fill the knowledge gap), some critical definition of how the GWP is estimated from field data, and some arguments to highlight how GWP maps contribute to a better management of groundwater.
Thus I recommend the manuscript to be reconsidered after the authors have addressed these major aspects.
More details below:
Can you summarize in the abstract what kind of input data (factors) and expected relationships with GWP are used to build the GWP maps?
Line 34: ‘Groundwater Potentially’- do you mean Groundwater Potential ? same remark throughout the text
Line 35: the GWP zone – not sure what is means - do you mean the GWP over an area of interest or GWP maps? Keep a consistent denomination through the manuscript.
Line 35 ‘teen’ – ten ?
Intro
More details are needed to describe which data and relationships are usually used to build GWP maps. This is worth a whole paragraph or a good extension of the second one.
Line 86: ‘Adopting an inventive approach’ could be removed as the present research is an application of existing methods.
Line 93: ‘allowing comparison with a previous study’ – can you be more precise (geographically and method used) and give a reference?
A paragraph summarizing previous studies and their limitations or the remaining knowledge gap at the site is missing, as well as some explanation as why is this specific site so important. This is somehow a bit of justification that is missing, with respect to what is announced in the last paragraph of the introduction. Do you use additional input data or factors? Additional GWP models? Additional validation data?
Material and methods
Lines 111,124,132,154,158,178,228,276,304,307,310, and many more: (Error! Reference source not found.b)
Line 114: not very clear, can you reformulate? I would use the terms spatial variability, temporal variability, difference in average rainfall.
Line 116: not very clear of what you refer to when talking about daily variation of temperature and then monthly average.
Figure 1 and in the text: By ‘withdrawal and non-withdrawal location’ do you mean production and observation boreholes?
Figure 2: the resolution is not good enough to read the text (which is too small for an A4 page); the A-A’ line on fig 2a does not seem properly located with respect to city location on 2c (in the middle).
Line 153: It should be summarized here and the different factor used should be defined as well. It is fine to refer to publications for more details, but the source of the data and the main principles of the methods (kind of inputs, outputs, training set, testing set, prediction set, validation criteria and model comparison and ranking) should be summarized here to give a good overview.
Figure 3: acronyms for the different factors should be defined prior to introducing the figure (or indicate in the caption in which table or sections they are defined).
Line 175-177: how do you define (non-)water withdrawal points? How do you relate them to GWP? Is GWP a continuous or a categorical variable? From my understanding after reading further, GWP is a continuous variable but that is uses as a categorical one for training and validation as set to 0 (non-water withdrawal) or 1 (water withdrawal). Is this correct? It should be clarified in the manuscript. On which quantitative criteria do you rely to you classify a point in one or the other category?
Lines 181 and 186: 20%/80% - do you mean 20% training and 80% testing?
Line 199 , and Figure 4: by closeness, do you mean distance?
Figure 4: increase the resolution. Some maps (all curvatures + convergence index) seems rather homogeneous – is it a problem of resolution or colour scale?
Line 254 eq 2 and 3: aren’t some j missing?
Lines 255+: while it is indicated how Tolj and VIFj are used to filter out some factors, how are MI and NFRn used?
Line 260: which analysis? Not sure to understand the link between the current paragraph and the previous one. Can you clarify?
Line 271: provide a reference to the Jenks natural break technique.
Line 273: from 2.3.3, which factors have finally been filtered out / selected?
Table 2: can you recall what the model acronym stands for in the description? E.g. ’Multi-layer perceptron is a fully…’
Line 282 ‘reinking’ – ranking
Results
Line 373: what is the difference between the prediction and the success rate? It should be defined at the end of section 2. What information does it bring?
I do not understand the computation of performance indicators on the training set. We cannot use the data both times for training and for validation. This is usually why the data is separated into a training set and a validation set, isn’t it?
Discussion
Line 456-458: can you reformulate? (Missing verb and subject)
What are your recommendations following the ranking? Could the methods be used jointly to better characterize the predictive uncertainty of GWP? Are some methods not good enough and should they be disregarded for GWP prediction in this study area?
Are there some specific characteristics such as some factor or spatial correlation for erroneous predictions across the different methods? In other words: is there some kind of identifiable pattern in the prediction errors?
What are the practical implications of the obtained GWP maps for groundwater resource management?
Author Response
Response to Reviewer 1 Comments
We appreciate Reviewer 1's valuable suggestions and comments, and we corrected the paper to reflect the improvements suggested. The answer to each comment is given below point-by-point (in red).
Point 1: Can you summarize in the abstract what kind of input data (factors) and expected relationships with GWP are used to build the GWP maps?
Response 1: New explanations have been added in the abstract to explain input data type and its relationship with GWP.
Point 2: Line 34: ‘Groundwater Potentially’- do you mean Groundwater Potential ? same remark throughout the text
Response 2: Thank you for your remark, we apologize for this typo.
Point 3: Line 35: the GWP zone – not sure what is means - do you mean the GWP over an area of interest or GWP maps? Keep a consistent denomination through the manuscript.
Response 3: This comment was reviewed and considered; a description of the GWP has been added.
Point 4: Line 35 ‘teen’ – ten ?
Response 4: Thank you for your remark, we apologize for this typo.
Intro
Point 5: More details are needed to describe which data and relationships are usually used to build GWP maps. This is worth a whole paragraph or a good extension of the second one.
Response 5: A new paragraph has been added to the introduction, in which we have included the indicators and methods commonly used to develop a GWP map from previous studies.
Point 6: Line 86: ‘Adopting an inventive approach’ could be removed as the present research is an application of existing methods.
Response 6: Thanks for your comment, we have edited the sentences.
Point 7: Line 93: ‘allowing comparison with a previous study’ – can you be more precise (geographically and method used) and give a reference?
Response 7: For the comparison of the individual models, new explanations have been added with references.
Point 8: A paragraph summarizing previous studies and their limitations or the remaining knowledge gap at the site is missing, as well as some explanation as why is this specific site so important. This is somehow a bit of justification that is missing, with respect to what is announced in the last paragraph of the introduction. Do you use additional input data or factors? Additional GWP models? Additional validation data?
Response 8: Please see the new paragraph added in the introduction. we have included basic information about the importance of the studied area.
Material and methods
Point 9: Lines 111,124,132,154,158,178,228,276,304,307,310, and many more: (Error! Reference source not found.b)
Response 9: We appreciate your remark, the errors in the reference source are corrected in the text.
Point 10: Line 114: not very clear, can you reformulate? I would use the terms spatial variability, temporal variability, difference in average rainfall.
Response 10: Thanks for your comment, we have rephrased the sentence.
Point 11: Line 116: not very clear of what you refer to when talking about daily variation of temperature and then monthly average.
Response 11: Thanks for your comment, we have edited the sentences.
Point 12: Figure 1 and in the text: By ‘withdrawal and non-withdrawal location’ do you mean production and observation boreholes?
Response 12: The withdrawal and non-withdrawal location’ is the location where there is no groundwater withdrawal. Please see the added explanation and the references in Section 2.3.1, "Inventory of Water Withdrawal Points."
Point 13: Figure 2: the resolution is not good enough to read the text (which is too small for an A4 page); the A-A’ line on fig 2a does not seem properly located with respect to city location on 2c (in the middle).
Response 13: Thanks for your comment, we have edited the Figure.
Point 14: Line 153: It should be summarized here and the different factor used should be defined as well. It is fine to refer to publications for more details, but the source of the data and the main principles of the methods (kind of inputs, outputs, training set, testing set, prediction set, validation criteria and model comparison and ranking) should be summarized here to give a good overview.
Response 14: For the Methodology section, new explanations and overall information on the used methods have been added. Please see the new information’s added in the methodology section.
Point 15: Figure 3: acronyms for the different factors should be defined prior to introducing the figure (or indicate in the caption in which table or sections they are defined).
Response 15: Thanks for your comment, we have defined the acronyms in text above the figure 3.
Point 16: Line 175-177: how do you define (non-)water withdrawal points? How do you relate them to GWP? Is GWP a continuous or a categorical variable? From my understanding after reading further, GWP is a continuous variable but that is uses as a categorical one for training and validation as set to 0 (non-water withdrawal) or 1 (water withdrawal). Is this correct? It should be clarified in the manuscript. On which quantitative criteria do you rely to you classify a point in one or the other category?
Response 16: For more explanation about the input data and GWP, please see the additional information and references in the text.
Point 17: Lines 181 and 186: 20%/80% - do you mean 20% training and 80% testing?
Response 17: Thank you for your remark, we apologize this is a typo, we have corrected it.
Point 18: Line 199 , and Figure 4: by closeness, do you mean distance?
Response18: For the closeness is defined on the text “closeness (i.e., distance to stream)
Point 19: Figure 4: increase the resolution. Some maps (all curvatures + convergence index) seems rather homogeneous – is it a problem of resolution or colour scale?
Response 19: Thanks for your comment, we have edited the Figure.
Point 20: Line 254 eq 2 and 3: aren’t some j missing?
Response 20: We appreciate your remark, we have added the missing j in the equation.
Point 21 Lines 255+: while it is indicated how Tolj and VIFj are used to filter out some factors, how are MI and NFRn used?
Response 21: In this section some explanation has been added. For the MI, it is indicated that negative values of MI signifying that the factor has no effect on the GWP. Therefore, this factor has to be eliminated. For the NFRn is the standarezed value of the FR, which represent the relationship between the water withdrawal locations (i.e., GWP) and the feature. It is used to unify the data type in numeric value.
Point 22: Line 260: which analysis? Not sure to understand the link between the current paragraph and the previous one. Can you clarify?
Response 22: Thanks for your comment, we have clarified the sentence.
Point 23: Line 271: provide a reference to the Jenks natural break technique.
Response 23: We appreciate your remark, we have provided the reference.
Point 24: Line 273: from 2.3.3, which factors have finally been filtered out / selected?
Response 24: Please see results in Section 3.1. GWP Influencing Factors Selection. “among the twenty-three factors the TRI, the LS and the slope were removed”
Point 25: Table 2: can you recall what the model acronym stands for in the description? E.g. ’Multi-layer perceptron is a fully…’
Response 25: We appreciate your remark, we have added models’ titles.
Point 26 Line 282 ‘reinking’ – ranking*
Response 26: Thank you for your remark, we apologize this is a typo, we have modified it.
Results
Point 27: Line 373: what is the difference between the prediction and the success rate? It should be defined at the end of section 2. What information does it bring?
Response 27: Thank you for your remark, please see the additional information in the text. The success rate reflects the accuracy of the model fit to the observed GWP. The prediction rate shows how well the model predicts the GWP.
Point 28: I do not understand the computation of performance indicators on the training set. We cannot use the data both times for training and for validation. This is usually why the data is separated into a training set and a validation set, isn’t it?
Response 28: Thank you for your remark, please see the additional information in the text. Computation of performance indicators on booth training and testing set it is widely applied in the previous study. It gives additional detail to the study.
Discussion
Point 29: Line 456-458: can you reformulate? (Missing verb and subject)
Response 29: Thanks for your comment, we have corrected the sentences.
Point 30: What are your recommendations following the ranking? Could the methods be used jointly to better characterize the predictive uncertainty of GWP? Are some methods not good enough and should they be disregarded for GWP prediction in this study area?
Response 30: Recommendations about the used models has been added in the discussion. Please see the added sentences in section 4.2. Models’ Performance and Periodization.
Point 31: Are there some specific characteristics such as some factor or spatial correlation for erroneous predictions across the different methods? In other words: is there some kind of identifiable pattern in the prediction errors?
Response 31: Thank you for your comment, please see the additional information in the text. in section 4.2. Models’ Performance and Periodization.
Point 32: What are the practical implications of the obtained GWP maps for groundwater resource management?
Response 32: Recommendations about the practical implications of the obtained GWP maps has been added in the discussion. Please see the added paragraph in section 4.3. Effectiveness of the Used Methodology.

Reviewer 2 Report
The paper is presenting the application of AI methods and in my opinion, it is an interesting subject for readers. The comparison of several AI methods and their application on a real scenario, makes it more important.
The paper needs minor edits in English but generally, it is acceptable.
It is not explained how the lithology map was converted to numbers to be used in the models. The map in the paper shows only alphabets representing the lithology.
There are many "Error! Reference source not found" through out the text. Please correct them.
Author Response
Response to Reviewer 2 Comments
We appreciate Reviewer 2's valuable suggestions and comments, and we corrected the paper to reflect the improvements suggested. The answer to each comment is given below point-by-point (in red).
Point 1: It is not explained how the lithology map was converted to numbers to be used in the models. The map in the paper shows only alphabets representing the lithology.
Response 1: The Lithology map and all the elaborated GWP influencing factors maps have been converted to numeric data using the NFR method (Standardized FR) it represent the relation between GWP and the Sub-Classes of the Factors. Please see the added explanation in Section “2.3.3. GWP Influencing Factors Optimized Selection Analysis”
Point 2: There are many "Error! Reference source not found" through out the text. Please correct them.
Response 2: We appreciate your remark, the errors in the reference source are corrected in the text.

Reviewer 3 Report
General comments
Robust research that needs conceptual explanations. Please, follow my guidelines to fix the issues
Specific comments
Line 35. “Ten” not “teen”
Line 51 - onwards. Explain better in the introduction the concept of GWP from a conceptual point of view
Lines 52-54. “Water stress is often frequent and widespread, therefore with a quasi-absence of surface water, groundwater represents the principal water supply source in those areas”. Add recent review paper from Sustainability that treats of surface/groundwater interaction and water scarcity:
- Medici, G. and Langman, J.B., 2022. Pathways and Estimate of Aquifer Recharge in a Flood Basalt Terrain; A Review from the South Fork Palouse River Basin (Columbia River Plateau, USA). Sustainability, 14(18), p.11349.
Lines 58-60. Regular quantitative and qualitative monitoring of groundwater constitutes an essential step toward its management, passing through its identification”. Sentence not backed up by references, insert relevant literature on the topic:
- Manna, F., J. Kennel, and B. L. Parker. "Understanding mechanisms of recharge through fractured sandstone using high-frequency water-level-response data. Hydrogeology Journal 30, no. 5 (2022): 1599-1618.
Line 124. Remove the error when you reference the figure. Here, and throughout the manuscript
Lines 109-149. Add the details on the presence of faults with links on groundwater flow. You introduce a cross-section with thrusts without providing sufficient detail
Line 215. Insert much more detail on fracture map, do you mean fault maps?
Line 334. How much shallow in terms of meters? Provide definition of shallow and deep in the manuscript
Line 422. The definition of “shallow models” in unclear in an hydrogeological manuscript. Please, define it to avoid misunderstandings
Line 438. “Influence of the geology”. Note that, I have asked adding geological details above
Line 439. Explain how the geology influences recharge processes, and storage. Any effects of faulting, or karstification? What about thickness of the quaternary cover? Later, you mention lithological effects but without providing an explanation
Line 448. Add age of the limestones. Jurassic?
Line 471. “As it” repeat the subject. Avoid starting a new sentence with “it”. Apply the correction throughout the manuscript.
Line 477. “Recent research”. Please, provide literature on the topic
Line 517. Can your research be also applied in non-arid settings? My impression is yes, please highlight this point to bring the impact out of your research
Line 534. Insert relevant references that I have suggested
Figures and tables
You need to remove the errors when you reference the figures in manuscript
Author Response
Response to Reviewer 3 Comments
We appreciate Reviewer 3's valuable suggestions and comments, and we corrected the paper to reflect the improvements suggested. The answer to each comment is given below point-by-point (in red).
Point 1: Line 35. “Ten” not “teen”
Response 1:Thank you for your remark, we apologize for this typo.
Point 2: Line 51 - onwards. Explain better in the introduction the concept of GWP from a conceptual point of view
Response 2: Thank you for your remark, we have outlined the concept of the GWP according to the literature. please see the second paragraph in the introduction
Point 3: Lines 52-54. “Water stress is often frequent and widespread, therefore with a quasi-absence of surface water, groundwater represents the principal water supply source in those areas”. Add recent review paper from Sustainability that treats of surface/groundwater interaction and water scarcity:
- Medici, G. and Langman, J.B., 2022. Pathways and Estimate of Aquifer Recharge in a Flood Basalt Terrain; A Review from the South Fork Palouse River Basin (Columbia River Plateau, USA). Sustainability, 14(18), p.11349.
Response 3: We appreciate your remark, references have been added.
Point 4: Lines 58-60. Regular quantitative and qualitative monitoring of groundwater constitutes an essential step toward its management, passing through its identification”. Sentence not backed up by references, insert relevant literature on the topic:
- Manna, F., J. Kennel, and B. L. Parker. "Understanding mechanisms of recharge through fractured sandstone using high-frequency water-level-response data. Hydrogeology Journal 30, no. 5 (2022): 1599-1618.
Response 4: We appreciate your remark, references have been added
Point 5: Line 124. Remove the error when you reference the figure. Here, and throughout the manuscript
Response 5: Thank you for your remark, the errors in the reference source are corrected in the text.
Point 6: Lines 109-149. Add the details on the presence of faults with links on groundwater flow. You introduce a cross-section with thrusts without providing sufficient
Response 6: We appreciate your remark, for more explanation, please see the additional detail and information in the text.
Point 7: Line 215. Insert much more detail on fracture map, do you mean fault maps?
Response 7: For more explanation about fracture map, please see the additional information and detail in the text.
Point 8: Line 334. How much shallow in terms of meters? Provide definition of shallow and deep in the manuscript
Response 8: We appreciate your remarks, shallow and deep are two different type of the Machine learning algorithms. For more details, please see Table 2.
Point 9: Line 422. The definition of “shallow models” in unclear in a hydrogeological manuscript. Please, define it to avoid misunderstandings.
Response 9: We appreciate your remarks; we have added de definition of Shallow models as it is described by Hongyu Liu 2019. please see the fifth paragraph in the introduction.
Point 10: Line 438. “Influence of the geology”. Note that, I have asked adding geological details above.
Response 10: Thank you for your comment, please see the additional information in the text. in section “4.1. Importance of GWP Influencing Factors”.
Point 11: Line 439. Explain how the geology influences recharge processes, and storage. Any effects of faulting, or karstification? What about thickness of the quaternary cover? Later, you mention lithological effects but without providing an explanation
Response 11: Thank you for your questions, please see the additional information and references in the text. In fact, areas with higher porosity (rocks permeability and/or fractures’) permit water infiltration, and provide pathways for water to flow into the subsurface.
Point 12: Line 448. Add age of the limestones. Jurassic?
Response 12: For the age of the limistones please see the modified text, it is noted “The Jurassic rocks are composed of limestone and dolostone (Liassic-Dogger)”
Point 13: Line 471. “As it” repeat the subject. Avoid starting a new sentence with “it”. Apply the correction throughout the manuscript.
Response 13: Thanks for your comment, we have rephrased the sentences.
Point 14: Line 477. “Recent research”. Please, provide literature on the topic
Response 14: Thanks for your comment, we have added relevant literature on the topic
Point 15: Line 517. Can your research be also applied in non-arid settings? My impression is yes, please highlight this point to bring the impact out of your research
Response 15: We appreciate your remarks; we have emphasized this point in the conclusion.
Point 16: Line 534. Insert relevant references that I have suggested
Response 16: thank you for your suggestion, the references have been added.
Figures and tables
Point 17: You need to remove the errors when you reference the figures in manuscript
Response 17: Thank you for your remark, the errors in the reference source are corrected in the text.
